# PAN-cancer analysis of S-phase enriched lncRNAs identifies oncogenic drivers and biomarkers

Mohamad Moustafa Ali [1], Vijay Suresh Akhade[1], Subazini Thankaswamy Kosalai[1], Santhilal Subhash [1], Luisa Statello[1], Matthieu Meryet-Figuiere[1], Jonas Abrahamsson[2], Tanmoy Mondal[1] & Chandrasekhar Kanduri[1]

Despite improvement in our understanding of long noncoding RNAs (lncRNAs) role in cancer, efforts to find clinically relevant cancer-associated lncRNAs are still lacking. Here, using nascent RNA capture sequencing, we identify 1145 temporally expressed S-phase-enriched lncRNAs. Among these, 570 lncRNAs show significant differential expression in at least one tumor type across TCGA data sets. Systematic clinical investigation of 14 Pan-Cancer data sets identified 633 independent prognostic markers. Silencing of the top differentially expressed and clinically relevant S-phase-enriched lncRNAs in several cancer models affects crucial cancer cell hallmarks. Mechanistic investigations on *SCAT7* in multiple cancer types reveal that it interacts with hnRNPK/YBX1 complex and affects cancer cell hallmarks through the regulation of FGF/FGFR and its downstream PI3K/AKT and MAPK pathways. We also implement a LNA-antisense oligo-based strategy to treat cancer cell line and patient-derived tumor (PDX) xenografts. Thus, this study provides a comprehensive list of lncRNA-based oncogenic drivers with potential prognostic value.

[1] Department of Medical Biochemistry and Cell Biology, Institute of Biomedicine, Sahlgrenska Academy, University of Gothenburg, Gothenburg 40530, Sweden. [2] Department of Pediatrics, Institution for Clinical Sciences, Sahlgrenska Academy, University of Gothenburg, Gothenburg 40530, Sweden. Mohamad Moustafa Ali, Vijay Suresh Akhade, Subazini Thankaswamy Kosalai and Santhilal Subhash contributed equally to this work. Correspondence and requests for materials should be addressed to C.K. (email: kanduri.chandrasekhar@gu.se)

Recent ultra-high-throughput transcriptome sequencing data suggest that nearly two-thirds of the human genome is pervasively transcribed, giving rise to more than double the number of lncRNAs compared to protein-coding RNAs[1]. It is evident from studies that lncRNAs do not represent transcriptional noise, but they participate in diverse biological functions with implications in development, differentiation, and disease[2–4]. lncRNAs operate at the transcriptional and post-transcriptional level to control gene expression in a spatiotemporal fashion[5]. Considering the influence of lncRNAs in a wide-range of biological processes, and that a significant portion of disease-associated single-nucleotide polymorphisms (SNPs) map to lncRNA loci[6], one could expect a greater role for lncRNAs in human disease. Recent investigations have also implicated numerous lncRNAs in cancer progression and metastatic dissemination[7–10]. Despite the increased number of studies addressing the role of lncRNAs in cancer, our understanding of lncRNAs in cancer initiation and progression as well as their relevance to clinical prognosis is in its infancy. Hence large-scale lncRNA-based functional screens, coupled with clinical investigations, are required to understand their roles in cancer progression and their use for diagnostic and prognostic purposes.

The ability to sustain chronic proliferation is considered one of the hallmarks of cancer[11]. The cell division cycle plays a central role in the control of normal and chronic cell proliferation via responding to cell cycle regulators such as cyclins and cyclin-dependent kinases (CDKs), growth factors and repressors, tumor suppressors and oncogenes[11]. Cell division comprises two phases: the DNA synthesis (S) phase and the mitotic (M) phase. The pathways controlling accurate DNA replication during S-phase are critical for normal cell proliferation, as replication errors could result in abnormal cell proliferation[12,13]. Thus, characterization of molecules that control S-phase progression would be of immense importance in understanding the role of S-phase in normal and disease conditions.

The role of lncRNAs in S-phase regulation is still poorly understood in comparison to protein-coding genes. Accumulated knowledge has so far been unable to explain the precise molecular pathways that control cell proliferation in normal and cancer conditions[11,14]. Thus, exploring the functional role of lncRNAs in cell cycle progression may provide insights into hitherto unanswered questions about how cell division is controlled in normal and cancer cells. Consistent with this, several investigations have shown that lncRNAs regulate cell cycle progression through controlling the expression of critical cell cycle regulators such as cyclins, CDKs, CDK inhibitors, and other factors[15–17]. In addition, a recent investigation implicated a subset of lncRNAs from 56 well-known cell cycle-linked loci, showing periodic expression patterns across cell cycle phases, in cell cycle regulation[18]. However, a comprehensive characterization of lncRNAs showing temporal expression during S-phase and their functional link to cell cycle regulation, cancer progression, and their use in prognosis has not been investigated.

In this study, we address two fundamental propositions that can enhance the current comprehension of the critical roles of lncRNAs in cancer development and progression. First, we study the contribution of S-phase-specific lncRNAs to cell cycle progression and other important cancer-associated hallmarks. Second, we investigate the capability of S-phase-specific lncRNAs to act as independent prognostic biomarkers in different cancers. To address these aims, we employ nascent RNA-seq during S-phase progression and generate a resource of lncRNA-based oncogenic cancer drivers that can be used as prognostic markers in the risk assessment as well as potential therapeutic regimens in the treatment of cancer.

## Results

**Identification of temporally expressed lncRNA across S-phase.** We have previously reported an optimized nascent RNA capture assay for detecting temporally expressed S-phase RNAs in HeLa cells[19]. Compared to the standard steady-state RNA isolation from various S-phase stages, this method detects accurate expression timing of the analyzed transcripts[19]. Briefly, HeLa cells were synchronized at the G1/S boundary using thymidine and hydroxyurea. Subsequently, the G1/S block was released and the cells were allowed to proceed in the presence of 5-ethynyl uridine (EtU) in order to specifically label the newly synthesized nascent RNA populations across different S-phase time points. To identify S-phase-specific lncRNAs, EtU labeled and steady-state (unlabeled) RNAs were individually collected at different time points spanning S-phase (T1, T2, and T3) in addition to RNA from unsynchronized cells (Fig. 1a). Following strand-specific library preparation, all RNA samples were subjected to deep sequencing (Supplementary Data 1). Principal component analysis of whole RNA, noncoding RNAs, and lncRNA expression profiles revealed that EtU-labeled RNAs had a better separation across different time points of S-phase compared to unlabeled samples (Fig. 1b). Based on the enrichment over unsynchronized samples, we identified 1734 and 1674 lncRNAs in EtU labeled and unlabeled samples, respectively. Further, to unveil the dynamically expressed S-phase lncRNAs, we performed Short Time-series Expression Miner (STEM) clustering[20]. We obtained 1145 lncRNAs with four significant temporal expression patterns, and 937 lncRNAs with two significant temporal patterns, in EtU labeled (Fig. 1c; Supplementary Data 1) and unlabeled samples (Supplementary Fig. 1a and Supplementary Data 1), respectively. Both EtU labeled and unlabeled samples shared 394 temporally expressed lncRNAs (Fig. 1c; Supplementary Fig. 1b). We next investigated the enrichment of E2F1 transcription factor binding sites at the promoters (±2 kb from transcription start site (TSS)) of temporally expressed lncRNAs and found that 72 EtU labeled and 67 unlabeled lncRNA promoters contain E2F1 binding sites (Supplementary Data 1). We randomly selected dynamically expressed S-phase lncRNAs and validated their temporal expression patterns across different cell cycle phases (G1, S, and G2/M) in EtU labeled and unlabeled samples (Supplementary Fig. 1c). To further confirm the specificity of this method in capturing temporally expressed lncRNAs during S-phase, we validated selected lncRNAs using a drug-free synchronization of HeLa cells using the serum starvation method (Supplementary Fig. 1d, e).

Since lncRNAs exert their actions via regulating protein-coding RNAs in cis and/or trans[21–24], we performed enrichment analysis using GeneSCF[24] for the neighboring protein-coding genes (within ±50 kb window of S-phase-specific lncRNAs). Interestingly, the neighboring protein-coding genes associated with EtU-labeled S-phase-specific lncRNAs demonstrated significant enrichment of molecular pathways related to cancer (Fig. 1d; Supplementary Data 1). Moreover, biological process analysis revealed an enrichment of transcriptional regulation, cell division, DNA repair, and cell migration processes. On the other hand, the protein-coding genes associated with unlabeled lncRNAs showed less enrichment of cancer-related pathways and cell division processes. These results indicate that nascent RNA enrichment via EtU labeling can efficiently distinguish temporally expressed lncRNAs during S-phase, which may be functionally engaged in vital cellular processes.

**S-phase lncRNAs show differential expression in cancers.** We next designed a workflow which integrated epigenomic, functional, and clinical approaches to address the functional

implications of EtU-labeled S-phase lncRNAs in multiple cancer types (Fig. 1e). Since, cell cycle perturbation is one of the hall-marks of cancer development[11], we addressed the functional connection between S-phase lncRNAs and cancer initiation and progression. We utilized the RNA-sequencing data of 6419 solid tumors and 701 normal tissue samples spanning 16 cancer types, from The Cancer Genome Atlas (TCGA)[25] (Supplementary

Data 2). Our analysis revealed that 570 out of 1145 S-phase-specific lncRNAs were differentially expressed in at least one cancer type with log-fold change ± 2 and false discovery rate (FDR) < 1E−004 (Methods section, Fig. 2a; Supplementary Data 2). Interestingly, nearly 73% of the differentially expressed lncRNAs show upregulation in corresponding cancer types. We found that 84 S-phase lncRNAs were differentially expressed

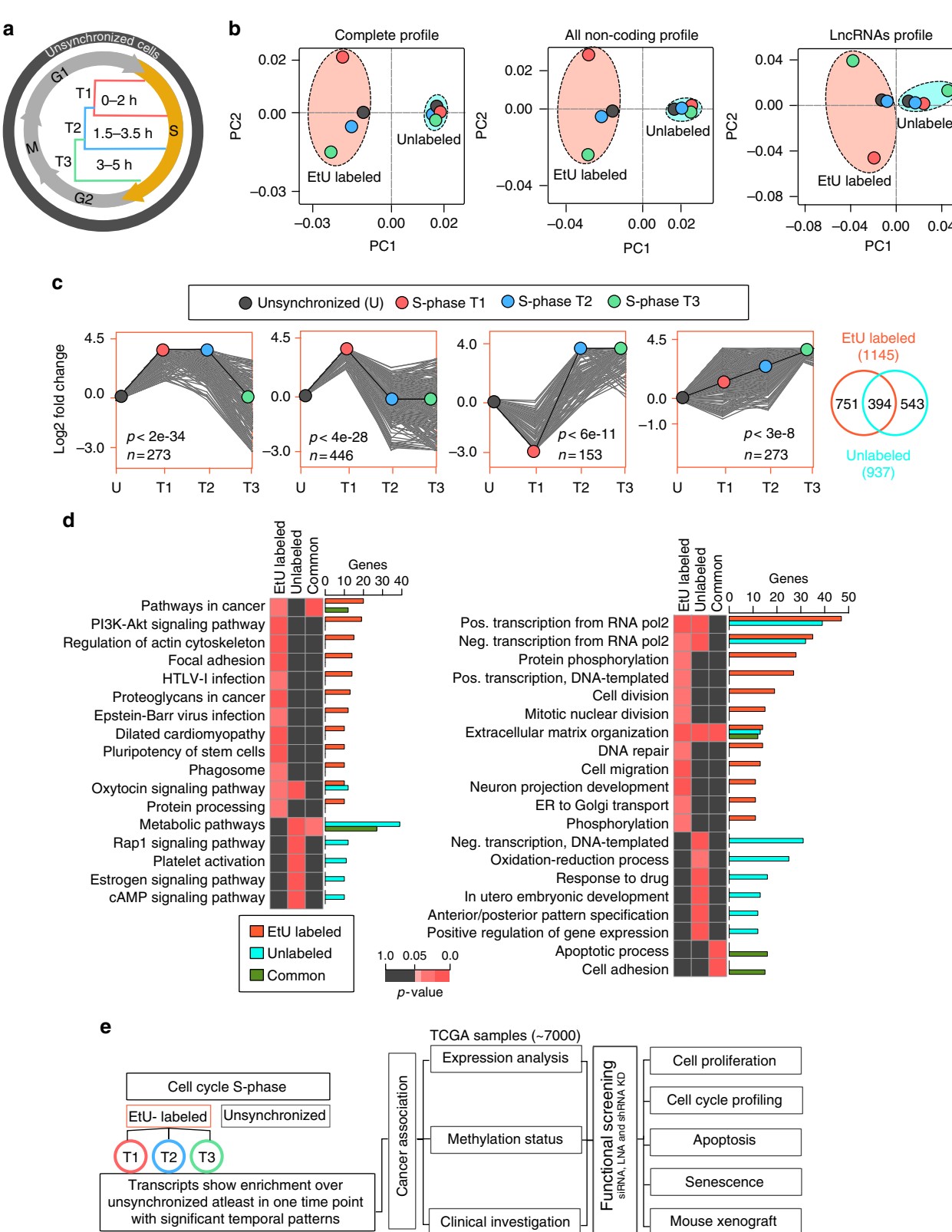

between normal and tumor tissues, in more than five cancer types, with four of them differentially expressed in 10 cancer types. Also, different cancer types originating from same tissues shared many differentially expressed S-phase lncRNAs (Fig. 2b). For instance, lung adenocarcinoma (LUAD) and lung squamous cell carcinoma (LUSC) share 98 differentially expressed S-phase lncRNAs. Likewise, kidney chromophobe (KICH), kidney papillary carcinoma (KIRP), and kidney clear cell carcinoma (KIRC) harbor 16 common differentially expressed S-phase lncRNAs. Since the latter cancers originate from distinct cell types[26,27], it remains to be seen whether these common and differentially expressed S-phase lncRNAs underpin the etiology of different kidney and lung cancers in a cell-independent manner. These observations indicate that temporally expressed lncRNAs show differential expression across multiple cancers and could serve as potential candidates for understanding their role in cancer development and progression.

**S-phase lncRNAs show epigenetic alterations**. We set out to address whether differential CpG methylation underlies the differential expression of S-phase lncRNAs seen in normal and tumor samples. We analyzed CpG methylation over their genomic regions in relation to gene expression. We utilized the processed CpG methylation data from the catalog of somatic mutations in cancers (COSMIC) for 12 TCGA cancer types (Infinium Human Methylation 450 beadchip platform; ~4000 samples, from international cancer genome consortium (ICGC) release 18)[28] (Supplementary Data 2). Differential methylation was considered significant only if a particular pattern (hypomethylation or hypermethylation) was supported by at least 10 patient samples and with null patients opposing the pattern (Methods section). CpG methylation analysis over 570 differentially expressed S-phase lncRNA promoters revealed 35 lncRNAs that exhibit differential methylation (Fig. 2c). Among these, 22 lncRNAs were hypomethylated with higher expression in tumors, whereas 13 lncRNAs were hypermethylated with lower expression in tumors. Since gene-body methylation has been shown to be positively correlated with gene expression[29], we analyzed gene-body methylation over S-phase lncRNAs and found 20 transcripts with differential methylation over the gene-body that correlated with gene expression (Supplementary Fig. 2). Out of 20 lncRNAs, 7 were hypermethylated and highly expressed, whereas 13 lncRNAs were hypomethylated with lower expression in tumors. Importantly, the promoter region of *RP11-152P17.2* was hypomethylated and highly expressed in HNSC, KIRP, and LIHC tumors compared to normal tissues (Supplementary Data 2).

**S-phase lncRNA act as independent prognostic markers**. We employed a systematic approach to identify the prognostic value of S-phase lncRNAs across 14 cancers. For each cancer type, we stratified the patients into high and low expression groups based on the expression levels of each lncRNA (Methods section). Then, we assessed the difference in survival using the Kaplan–Meier

(KM) method. We tested the randomness in survival prediction using receiver operating characteristic[7] curves and filtered candidates with an area under the curve (AUC) less than 0.5. To calculate the hazard ratios associated with the expression of each S-phase lncRNAs in each respective cancer type, we generated a multivariate Cox-regression model (Supplementary Data 2). The accuracy of the predictive multivariate prognostic models was investigated using Brier score-based prediction error curve over the reference model. Our systematic analysis based on dichotomization approach identified 520 S-phase lncRNAs that predict survival and also act as potential independent biomarkers at least in one cancer type (Fig. 2d, e; Supplementary Data 2). In addition, we identified 375 S-phase lncRNAs as continuous biomarkers in at least one cancer type (Supplementary Data 2) with a significant overlap of 262 S-phase lncRNAs with the dichotomization-based approach (Supplementary Fig. 2b and Supplementary Data 2). Notably, the multivariate models demonstrated that KIRC harbors the maximum number of S-phase lncRNA-based prognostic biomarkers. Of note, *RP11-59D5__B.2* is the top candidate in our clinical investigations and it acts as an independent prognostic indicator in five cancers: BLCA, KIRC, KIRP, STAD, and HNSC (Supplementary Fig. 2c-f and Supplementary Data 2). Taken together, a large proportion of S-phase lncRNAs show an independent prognostic capacity, in relation to clinical covariates, in different cancer types.

**SCATs modulate cell cycle progression and cell proliferation**. For further functional validation, we selected eight S-phase lncRNAs among the top candidates having a higher frequency of differential expression across TCGA data sets (Fig. 2a) which also independently predict the patients' survival in multiple cancers (Fig. 2d, Fig. 3a; Supplementary Fig. 3). Hereafter, we address the selected candidates as S-phase cancer-associated transcripts (SCATs). We depleted the SCATs using small-interfering RNA oligonucleotides (siRNAs) or locked nucleic acid (LNA) antisense oligonucleotides using HeLa cells as a model system (Supplementary Fig. 3i). We assessed the effect of SCATs downregulation on cell proliferation using the colorimetric MTT assay (Fig. 3b). SCATs depletion induced a significant effect on cell proliferation. Similarly, the impact of SCATs loss-of-function on cell cycle progression was assessed using flow cytometry (Fig. 3c). Loss of function of SCATs induced cell cycle perturbations, accompanied with an accumulation of cells at the G1 phase and a decrease in DNA synthesis. Furthermore, downregulation of SCATs induced cellular apoptosis, as indicated by a significant increase in caspase 3/7 activity (Fig. 3d).

Since some of the SCATs showed differential expression and predict survival in KIRC tumors, we used a KIRC cell line Caki-2, to further validate their functional role in cell homeostasis. First, we investigated the temporal expression patterns of *SCAT4, SCAT5, SCAT7,* and *SCAT8* during cell cycle progression in serum-starved Caki-2 cells and found their elevated expression during S-phase (Supplementary Fig. 3j). Then, we downregulated *SCAT4, SCAT7,* and *SCAT8* (Supplementary Fig. 3k) and

**Fig. 1** Identification of temporally expressed lncRNAs across S-phase using nascent RNA capture assay. **a** Cell cycle diagram showing Nascent RNA capture at three different time points (2, 3.5, and 5 h) of S-phase. **b** Principle component analysis (PCA) of expression profiles by considering complete profile (noncoding and protein-coding), all noncoding and only lncRNAs. **c** Time-series analysis of S-phase associated lncRNAs with twofold enrichment at least in one time point over the unsynchronized sample. The S-phase lncRNAs show four significant temporal patterns with STEM clustering. Venn diagram shows the overlap of lncRNAs enriched in EtU labeled and unlabeled samples. The *p*-values are obtained using permutation tests from STEM clustering. **d** Molecular pathway analysis (left) and gene ontology (right) enrichment analysis for nearby (<50 kb proximity) protein-coding genes to S-phase associated lncRNAs. The KEGG pathways or biological functions presented in the heatmaps show at least 10 genes with *p*-value < 0.05. The hypergeometric *p*-values are obtained from GeneSCF for pathway and GO enrichment analysis. **e** Workflow of the analysis for the identification of S-phase associated lncRNAs and its significance in different cancers

measured cell proliferation, cell cycle profile, and caspase activity. Consistent with the loss-of-function experiments in HeLa cells, depletion of the selected SCATs in Caki-2 cells altered cell proliferation, inhibited cell cycle progression, and induced apoptosis (Fig. 3e–g).

**SCAT1 and SCAT5 act as as independent prognostic factors.** Our clinical investigations revealed *CTD-2357A8.3* (*SCAT1*) and *LUCAT1* (*SCAT5*) as common independent prognostic biomarkers for lung and kidney-derived cancers, respectively (Fig. 2e). *SCAT1* is differentially expressed in 10 cancers (Fig. 4a),

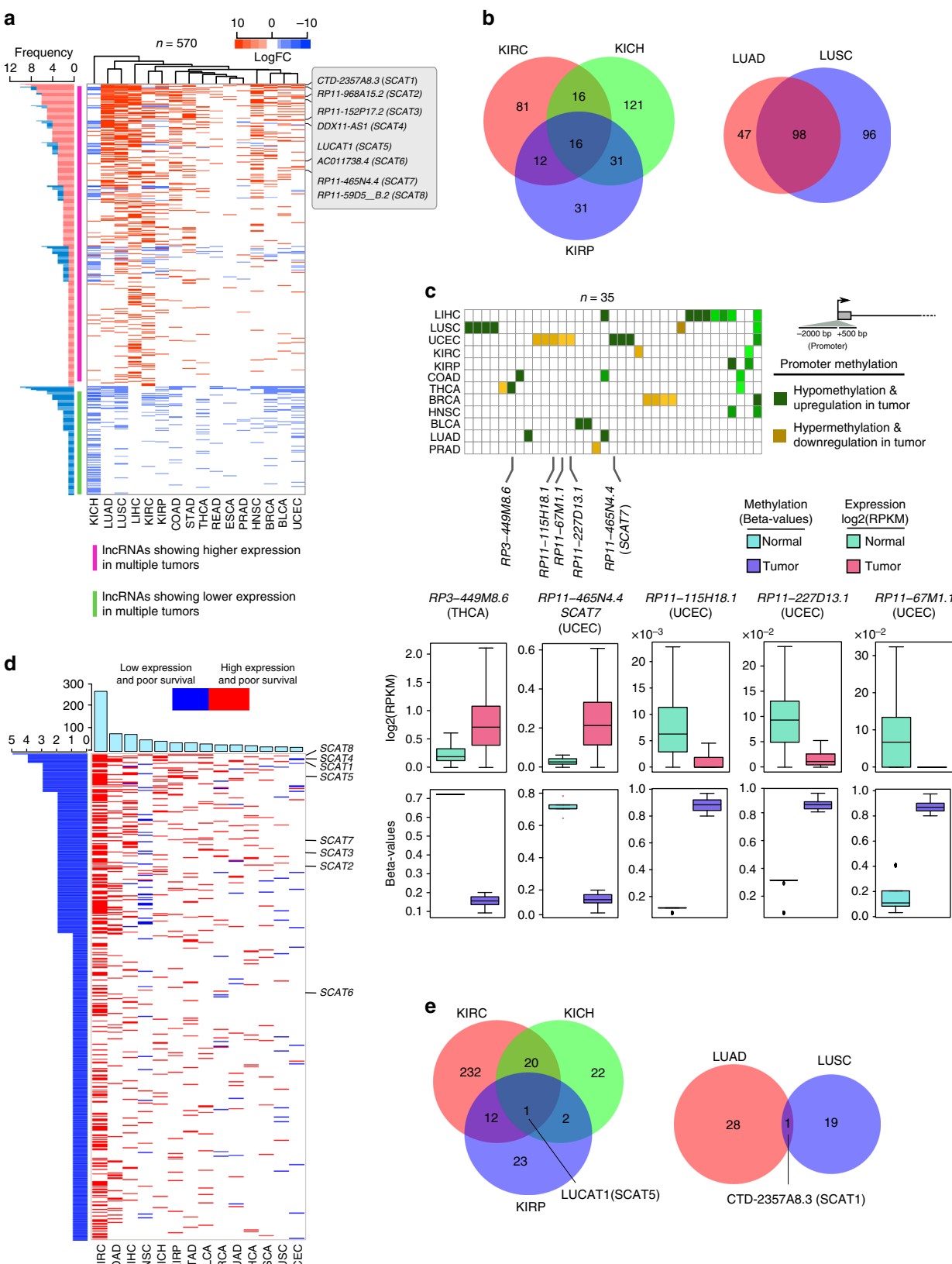

including LUAD and LUSC. Higher expression of *SCAT1* in both LUAD and LUSC correlates with the poor clinical outcome (Fig. 4b; Supplementary Fig. 4a-f). Additionally, *SCAT1* shows an elevated expression in the S-phase of synchronized A549 cells (Supplementary Fig. 4g). Consistent with the clinical data, downregulation of *SCAT1* in A549 cells exhibited a drastic inhibition of cell proliferation (Fig. 4c; Supplementary Fig. 4h), cell cycle arrest at G1 phase (Fig. 4d), and promoted cellular apoptosis (Fig. 4e), indicating its role in lung carcinogenesis. Similarly, *SCAT5*, which is differentially expressed in five cancers (Fig. 4f), acts as an independent prognosticator in all three types of kidney cancers (KIRP, KIRC, and KICH) (Figs. 3a and 4g; Supplementary Fig. 4i-n and Supplementary Data 2). Its higher expression predicts poor clinical outcome in all three cancers and similar to other SCATs, *SCAT5* expression is induced during S-phase in Caki-2 cells (Supplementary Fig. 3j). The depletion of *SCAT5* in Caki-2 cells led to decrease in cell proliferation, cell cycle arrest at G1 phase, and induces apoptosis (Fig. 4h-j; Supplementary Fig. 4o).

**SCAT7 modulates hallmarks of cancer across cell lines**. We chose *RP11-465N4.4* (*SCAT7*) to gain mechanistic insights into S-phase lncRNA induced cancer initiation and progression. Current annotations refer to *SCAT7* as a polyadenylated antisense lncRNA with two variants of 788 and 500 nucleotides. The 788 nucleotide variant spans two protein-coding genes (RNPEP and ELF3). Northern blot analysis also confirmed the presence of these two variants (Supplementary Fig. 5a). The noncoding capacity of *SCAT7* was confirmed using both CPAT probability and CPC score (Supplementary Fig. 5a). The sequence conservation analysis of *SCAT7*, as determined by phastCons score and phyloP, indicates no sequence homology among vertebrates (Supplementary Fig. 5b). *SCAT7* RNA copy number analysis of HeLa and A549 cells by droplet digital PCR (ddPCR) revealed 224 and 496 copies/ng RNA, respectively (A549 cells known to have >2500 copies *MALAT1* and *NEAT1*[30]).

*SCAT7* was upregulated in multiple cancers (BLCA, BRCA, KIRP, LIHC, LUAD, LUSC, PRAD, and UCEC) (Fig. 5a). Moreover, our detailed clinical investigations demonstrated its potential to independently predict clinical outcome in KIRC and COAD patients (Fig. 3a; Supplementary Fig. 3g and Supplementary Data 2). Furthermore, *SCAT7* showed significant differential methylation in UCEC patients (Fig. 2c; Supplementary Data 2). The temporal expression analysis of *SCAT7* during cell cycle progression indicated elevated levels in the S-phase of HeLa and Caki-2 cell lines (Supplementary Figs. 1c, d and 3j). Furthermore, qPCR expression analysis indicated that *SCAT7*

exhibits an elevated expression in LUAD cell lines (A549 and H2228) compared to normal lung cells (Supplementary Fig. 5c). Collectively, these results indicate that *SCAT7* could be an oncogenic lncRNA with a potential prognostic capability. Hence, we sought to investigate the functional role of *SCAT7* in the maintenance of cancer cell hallmarks in different cell lines representing multiple cancer types. To this end, both variants of *SCAT7* were downregulated in cell lines representing KIRC (Caki-2 and 786-O), LUAD (A549 and H2228), LIHC (HEPG2), and BRCA (MCF7) using siRNA or short hairpin RNA (shRNA). Downregulation of *SCAT7* variants in the knockdown cells in Northern blot analysis indicates the specificity of our RNAi knockdown experiments (Supplementary Fig. 5a). Downregulation of *SCAT7* in Caki-2, 786-O, A549, H2228, HepG2, and MCF7 cells affected cell proliferation (Fig. 5b; Supplementary Fig. 5d, e) and cell cycle progression, with preferential accumulation of cells at G1 phase (Fig. 5c; Supplementary Fig. 5f). Additionally, the knockdown of *SCAT7* led to decreased proliferation even in the non-cancerous HEK293 cells (Supplementary Fig. 5g). Furthermore, *SCAT7* depletion altered the S-phase progression in multiple cell lines as indicated by the nascent 5-ethynyl-2'-deoxyuridine analog (EdU) incorporation assay (Supplementary Fig. 5h). Notably, compared to other cell lines, the transient knockdown of *SCAT7* in H2228 cells induced a significant accumulation of cells at G2/M phase (Supplementary Fig. 5f).

We next focused on Caki-2, 786-O, and A549 stable KD cell lines to elucidate the functional role of *SCAT7* in apoptosis, cell viability, cell migration and invasion. *SCAT7* knockdown in Caki-2 and A549 cell lines increased caspase 3/7 activity (Fig. 5d) and affected their viability, migration, and invasion properties (Fig. 5e, f; Supplementary Fig. 5i-m). The invasion capacity was severely suppressed in Caki-2 and A549 knockdown cells, while the effect was very limited in the case of 786-O knockdown cells. Additionally, we investigated the anchorage-independent cell growth using soft agar colony formation assay for Caki-2, 786-O, and A549 knockdown cell lines 10 days after incubation (Fig. 5g; Supplementary Fig. 5n). The ability of knockdown cells to form anchorage-independent colonies was drastically reduced, and accordingly, the total number of colonies, as well as the area of each individual colony was decreased. In accordance with its role in restricting the cellular proliferation upon knockdown, overexpression of *SCAT7* increased cell proliferation by 2.25, 2.08, and 1.9 fold in HeLa, Caki-2, and A549 cells, respectively (Fig. 5h). However, we were unable to overexpress *SCAT7* in the 786-O cell line due to its inherited resistance to plasmid transfection. Collectively, our data demonstrate the critical role of *SCAT7* in regulating some of the most important cancer hallmarks in multiple cancer cell lines.

**Fig. 2** Characterization of S-phase lncRNAs as oncogenic drivers and independent prognostic markers using pan-cancer TCGA data sets. **a** Heatmap of 570 S-phase lncRNAs showing significant differential expression at least in one cancer type from TCGA. The significance was considered based on log-fold change ± 2 and FDR < 1E−004 (Benjamini–Hochberg's method). The highlighted lncRNAs are the ones selected for functional validation. The plot on the left side shows the frequency of individual lncRNAs differentially expressed across different cancer types. **b** Venn diagrams illustrating the overlap between common S-phase lncRNAs which are differentially expressed in different types of kidney and lung cancers. **c** Heatmap of S-phase lncRNAs with anti-correlative promoter methylation status to the differential expression levels in corresponding cancer. Hypomethylated promoter associated with higher expression and hypermethylation to less expression compared to normal. The highlighted lncRNAs are differentially methylated in more than 100 patients (samples) supporting the methylation pattern in corresponding cancer. Box plots of highlighted lncRNAs denoting the anti-correlative relation between promoter methylation and transcript expression. Box plots middle line shows the median, the box limits are 25th and 75th percentiles, whiskers are nearer quartile ± 1.5 times interquartile range and points beyond whiskers are the outliers. The *p*-values for the comparisons were obtained using Mann–Whitney test and corrected *p*-values using Family Wise Error Rate (FWER). The significant differentially methylated regions were filtered on the basis of FWER < 0.05. The above information for statistical analysis was extracted from COSMIC repository. **d** Heatmap showing the potential independent prognostic values of 520 S-phase lncRNAs based on dichotomization approach. The red color indicates higher expression of lncRNAs that predicts poor survival outcome. The blue color indicates lower expression of lncRNAs associated with poor survival in patients. **e** Venn diagrams indicating the numbers of S-phase lncRNAs that independently predict the survival outcome in different types of kidney and lung cancers

**Downregulation of *SCAT7* induces cell senescence**. Given that *SCAT7* knockdown induces cell cycle perturbations and deformed cell morphology (Supplementary Fig. 5o), we investigated the contribution of *SCAT7* in cell senescence. To this end, we used two fibroblast cell lines; BJ and TIG3, immortalized with B-RAF transformation. Transient downregulation of *SCAT7* using three siRNAs for 72 h induced senescence in both the cell lines as

indicated by β-galactosidase staining and upregulation of senescence markers like *p16*, *IL8*, and *p21* (Fig. 5i, j; Supplementary Fig. 5p). We also observed a spontaneous decrease in the expression levels of *SCAT7* during serial passaging of fibroblast cells mimicking the normal physiological aging (Fig. 5k). The immortalized BJ-BRAF fibroblasts express the activated form of mouse B-RAF (V600E) under the control of estrogen receptor

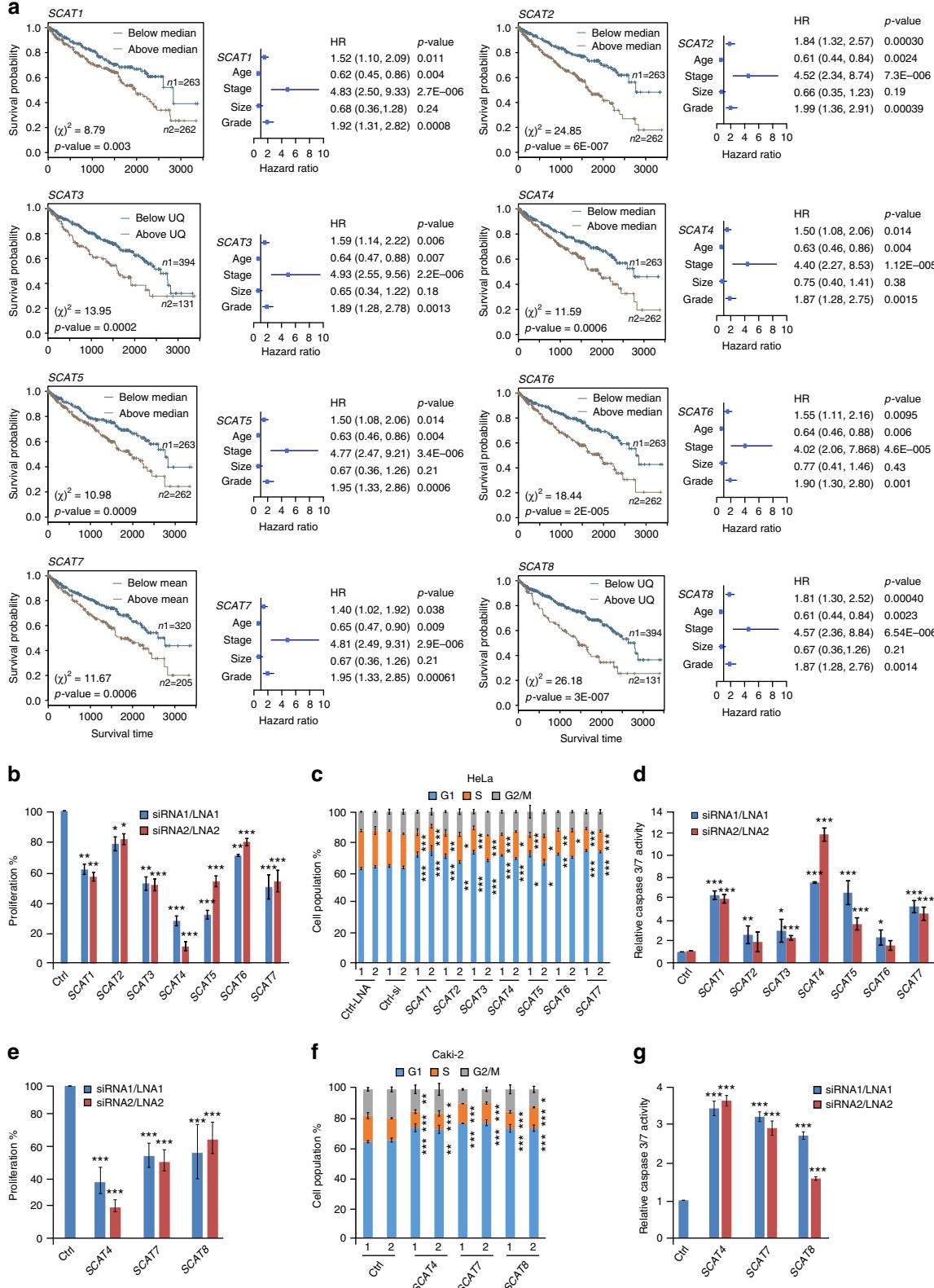

(ER:B-RAF)[31]. Thus, the addition of 4-hydroxytamoxifen (4-OHT) activates B-RAF and consequently senescence-associated signaling pathways[32,33]. We sought then to assess the transcriptional activity of *SCAT7* upon induction of premature senescence. Towards this, we treated BJ-BRAF cells with 200 nM of 4-HT for 72 h and checked the transcriptional activation of some senescence-associated markers (Supplementary Fig. 5q). Interestingly, the onset of premature senescence resulted in significant reduction of *SCAT7* expression (Fig. 5l). On the other hand, overexpression of *SCAT7* in BJ-BRAF fibroblasts increased their proliferation (Fig. 5m; Supplementary Fig. 5r) as well as transcriptional activity of key senescence markers (Supplementary Fig. 5s). Cells overexpressing *SCAT7* were able to bypass the tamoxifen-induced senescence, as indicated by less β-galactosidase activity and transcriptional repression of the key senescence markers *p16*, *p15*, and *IL8* (Fig. 5n-p). We next tested the hypothesis that *SCAT7* silencing in cancer cell lines may promote senescence. Toward this, we performed β-galactosidase staining on HeLa, A549 and Caki-2 cell lines upon *SCAT7* downregulation. Downregulation of *SCAT7* induced the senescence phenotype in both HeLa and A549 cells while Caki-2 cells remained unaffected (Fig. 5q). Therefore, our data demonstrate the crucial role of *SCAT7* in regulating cellular senescence, highlights the crosstalk between cell proliferation, cell cycle progression, and senescence.

**SCAT7 regulates FGF/FGFR, PI3K/AKT, and Ras/MAPK pathways**. To gain an insight into the *SCAT7*-mediated molecular pathways and cancer-related processes, we performed RNA sequencing of HeLa, Caki-2, and A549 cells upon *SCAT7* downregulation (Supplementary Fig. 6a). Subsequent functional enrichment analysis of RNA-seq data revealed that the depletion of *SCAT7* affected major signaling pathways and vital biological processes (Fig. 6a-c; Supplementary Data 3). For instance, FGF/FGFR and the downstream PI3K/AKT and Ras/MAPK pathways were largely affected while cell proliferation, cell adhesion, cellular senescence, cell migration, and apoptotic processes were also perturbed. We validated some of the dysregulated genes at the transcriptional and translational levels (Fig. 6d and Supplementary Fig. 6b) in the *SCAT7* downregulated cells. We detected a reduction in FGFR, p-AKT, and p-ERK1/2 levels in the *SCAT7* downregulated HeLa, Caki-2, and A549 cells. We also investigated the status of some of the well-known cell cycle master regulators; such as CCND1, CDK4, RB, and Phospho-Ser795 RB (p-RB), in *SCAT7* depleted cells (Fig. 6e). *CCND1* was downregulated at the transcriptional and translational level upon *SCAT7* knockdown while p-RB, but not RB, was downregulated at the protein level. Conversely, the CDK4 expression was neither affected at the transcriptional nor the translational levels. Since *SCAT7* overlaps two neighboring protein-coding genes *ELF3* and *RNPEP*, we investigated the effect of its downregulation on the

neighboring genes. *SCAT7* depletion, using different siRNAs, shRNAs and LNAs targeting the unique non-overlapping exon of *SCAT7*, did not affect the expression of neighboring genes in HeLa and Caki-2 cell lines. However, we found reduced *ELF3* expression only in A549 cells (Supplementary Data 3).

Based on RNA-seq analysis and subsequent validations across different cell lines, we hypothesized that *SCAT7* modulates some of the cancer hallmarks through the regulation of different FGF/FGFR members. For instance, *SCAT7* knockdown affected the mRNA levels of *FGFR2*, *FGFR3*, *FGF7*, and *FGF21* in HeLa cells while only *FGFR2* was downregulated in the Caki-2 and 786-O cells (Fig. 6f; Supplementary Fig. 6c). Similarly, *FGFR3* and *FGFR4* were downregulated upon *SCAT7* depletion in A549 cells (Fig. 6f). Conversely, overexpression of *SCAT7* restored the expression of *FGFR2* in HeLa and Caki-2 cells (Fig. 6g). Moreover, the expression pattern of different *FGFR* members matched *SCAT7* expression during cell cycle progression in the investigated cell lines (Supplementary Fig. 6d) and in tumor tissues derived from LUAD, KIRC, and UCEC (Supplementary Data 3). For instance, in HeLa cells, *SCAT7*, *FGFR2*, and *FGFR3* are expressed early in the S-phase. In the case of A549 cells, *SCAT7*, *FGFR3*, and *FGFR4* demonstrate a higher expression at G2 phase. To test our hypothesis that *SCAT7* executes its actions via FGF/FGFR signaling, *FGFR2* was depleted in HeLa and Caki-2 cells using two siRNAs, and *FGFR3* and *FGFR4*, separately, using esiRNAs in A549 cells (Supplementary Fig. 6e). Interestingly, the effects of the *FGFR2* depletion on cell cycle, cell proliferation, and vitality phenocopied the effects of the *SCAT7* depletion in the HeLa and Caki-2 cells (Fig. 6h, i; Supplementary Fig. 6f). Furthermore, *SCAT7*-induced cell proliferation was attenuated when *FGFR2* was silenced; indicating that *SCAT7*-induced cell proliferation is, in part, carried out by the FGF/FGFR2 signaling (Supplementary Fig. 6g). Though downregulation of both *FGFR3* and *FGFR4* in A459 cells led to a decrease in cell proliferation, only *FGFR3* downregulation affected cell cycle progression in A549 cells (Fig. 6h, i; Supplementary Fig. 6h, i). Collectively, our data strongly suggest the involvement of *SCAT7* in the modulation of cellular homeostasis and cell cycle progression through the regulation of the FGF/FGFR and the downstream PI3K/AKT and Ras/MAPK signaling pathways.

**SCAT7/hnRNPK/YBX1 complex regulates cancer cell hallmarks**. We next studied the mechanisms by which *SCAT7* regulates cancer progression via FGF/FGFR signaling in HeLa cells. We first determined the subcellular distribution of *SCAT7* and found it to be enriched in the nucleoplasmic and chromatin compartments (Supplementary Fig. 7a). Then, we performed chromatin oligo-affinity precipitation (ChOP) in UV crosslinked cells using biotinylated oligonucleotides to pull-down *SCAT7* and its interacting proteins. Using two independent biological replicates, we identified 96 proteins that were specifically enriched

**Fig. 3** The top clinically relevant S-phase lncRNAs regulate crucial cancer cell hallmarks. **a** Kaplan–Meier plots of *SCAT1–SCAT8* indicating overall survival of patients in KIRC. The higher expression of all SCATs is correlated with poor overall survival. The expression cut-off and the significance value for each SCAT are indicated in the plots. UQ represents upper quartile of the patients' expression levels. The Forest plots represent the multivariate models derived for each SCAT in combination with the significant clinical parameters. The hazard ratio (HR) using Cox proportional hazard analysis and the associated *p*-values were calculated using Wald statistics. **b** Proliferation capacity of HeLa cells as measured by MTT colorimetric assay 48 h post-silencing of SCATs using two different LNAs (*SCAT1*, *SCAT2*, *SCAT4*, *SCAT5*, and *SCAT6*) or siRNAs (*SCAT3* and *SCAT7*). Data are represented as percentage compared to cells transfected with respect to the negative control. No significant difference was observed between LNA-negative control and siRNA-negative control. **c** Cell cycle profiles of HeLa cells depleted with two different LNAs or siRNAs targeting the seven SCATs. **d** Estimation of the caspase 3/7 activities 48 h post-silencing of SCATs in HeLa cells. Data are expressed as fold change with respect to the corresponding negative controls. **e** MTT proliferation assay of Caki-2 (KIRC) cell line depleted with two independent LNAs (*SCAT4*, *SCAT8*) or siRNAs (*SCAT7*). **f** Cell cycle profiles of Caki-2 cells depleted with two different LNAs or siRNAs. **g** Estimation of the caspase 3/7 activities 48 h post-silencing of the corresponding SCAT in Caki-2 cells. Data in **b**–**g** are shown as mean ± SEM of three independent experiments. Significance levels were derived using unpaired two-tailed Students' *t*-test. (*$p \leq 0.05$; **$p = 0.01$–$0.001$; ***$p < 0.001$)

in both replicates of *SCAT7* ChOP (Fig. 7a; Supplementary Data 3). We selected two proteins, hnRNPK and YBX1, well-known for functions related to transcriptional regulation and cell cycle progression, for detailed mechanistic investigations[34,35]. We validated the interaction of hnRNPK and YBX1 proteins with *SCAT7* using RNA immunoprecipitation (RIP) assay (Fig. 7b) and ChOP pull-down followed by Western blot (Fig. 7c).

Next, we investigated the role of hnRNPK and YBX1 in the regulation of cell proliferation, cell cycle progression, *FGFR2*, and *FGFR3* expression, and pathways downstream to FGFRs in HeLa cells. Transient knockdown of hnRNPK or YBX1 by siRNAs (Supplementary Fig. 7b) led to a significant decrease in cell proliferation (Fig. 7d) and cell cycle arrest at the G1 phase (Fig. 7e). Also, *FGFR2* and *FGFR3* expression was substantially

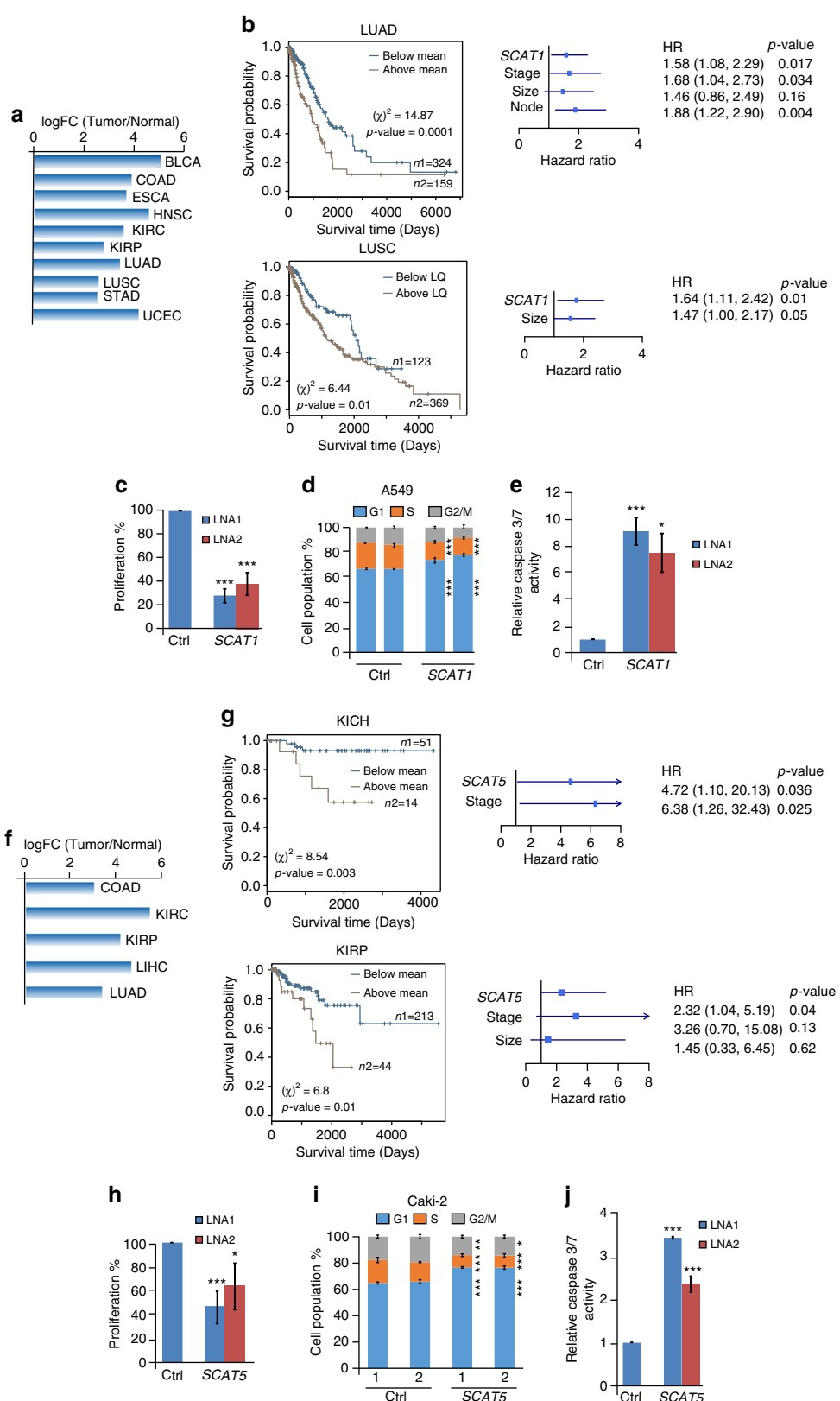

downregulated at the mRNA (Fig. 7f) and protein levels (Fig. 7g) in *hnRNPK* or *YBX1* knockdown cells. The activities of the MAPK and AKT pathways downstream to FGFRs were also reduced upon *hnRNPK* or *YBX1* knockdown (Fig. 7g). To understand the mechanism of *FGFR2* and *FGFR3* regulation by *SCAT7*/hnRNPK/YBX1 ribonucleoprotein (RNP) complex, we first checked the interaction of *SCAT7*/hnRNPK/YBX1 complex with mRNA and the proximal promoters of *FGFR2* and *FGFR3* genes. In our RIP assays with hnRNPK and YBX1, we failed to detect *FGFR2* or *FGFR3* mRNAs, indicating that *SCAT7* does not modulate *FGFR2* and *FGFR3* mRNA levels post-transcriptionally (Supplementary Fig. 7c). We then assessed the occupancy of *SCAT7* over the *FGFR2* and *FGFR3* promoter regions using ChOP-qPCR (Fig. 7h, i). Our analysis revealed a significant enrichment of *SCAT7* at the *FGFR2* (−250 to −750 bp relative to TSS (Fig. 7h) and *FGFR3* (−750 bp to −1 kb relative to TSS) (Fig. 7i) promoters. This observation led us to check whether *SCAT7* mediates the transcriptional activation of *FGFR2* and *FGFR3* via recruiting hnRNPK and YBX1 to the *FGFR2* and *FGFR3* promoters. We detected specific binding of hnRNPK and YBX1, as assessed by ChIP assay, at the *FGFR2* promoter (−250 to −500 bp relative to TSS) and this binding was substantially reduced upon *SCAT7* downregulation (Fig. 7j). Similarly, hnRNPK and YBX1 occupied the −500 bp to −1 kb promoter region of *FGFR3* and their binding was affected specifically at the −750 bp to −1 kb promoter region following *SCAT7* downregulation (Fig. 7k).

Further, we investigated the role of *SCAT7*/hnRNPK/YBX1 complex in A549 and Caki-2 cells for transcriptional regulation of *FGFR3* and *FGFR2*, respectively. We performed *SCAT7* ChOP pull-downs in A549 (Fig. 7l) and Caki-2 (Fig. 7o) cells followed by Western blotting. ChOP-Western indicated a specific interaction of *SCAT7* with both hnRNPK and YBX1 proteins in A549 cells (Fig. 7l), but only with hnRNPK in Caki-2 cells (Fig. 7o). ChOP-qPCR assays in A549 and Caki-2 cells revealed *SCAT7* occupancy at the promoter regions of *FGFR3* (−750 bp to −1 kb) and *FGFR2* (−250 bp to −500 bp), respectively (Fig. 7m, p). Similar to our observations in HeLa cells, we detected the occupancy of hnRNPK and YBX1 at the −750 bp to −1 kb *FGFR3* promoter region in A549 cells and their occupancy was reduced upon *SCAT7* downregulation (Fig. 7n). In Caki-2 cell line, hnRNPK was enriched at the −250 to −500 bp *FGFR2* promoter region and its occupancy was affected upon *SCAT7* downregulation (Fig. 7q). In addition, we observed a decrease in the RNA Polymerase II occupancy over the coding region of FGFRs in *SCAT7* depleted cells (Supplementary Fig. 7d). Taken together, these observations indicate that *SCAT7*/hnRNPK/YBX1 RNP plays a crucial role in the transcriptional activation of *FGFR2* and/or *FGFR3* in different cancer models.

**SCAT7 as potential therapeutic target in cancer treatment.** Our in vitro data, as well as the mechanistic studies, clearly demonstrate the oncogenic nature of *SCAT7* and its crucial role in promoting cancer-associated signaling pathways. Hence, we wanted to elucidate the role of *SCAT7* in malignant tumorigenesis in vivo. To this end, we generated two different xenograft models engrafted with either 786-O or A549 *SCAT7* stable knockdown cells. Eight weeks post-engraftment, both *SCAT7* depleted 786-O and A549 xenografts showed a significant decrease in growth parameters compared to control xenografts. Ki67 immunostaining of the dissected tumors confirmed the restricted proliferation capacity of *SCAT7* knockdown cells in vivo (Fig. 8a, b; Supplementary Fig. 8a, b).

We next tested the hypothesis that *SCAT7* can serve as a target for therapeutic intervention. Towards this, we engrafted A549 cells subcutaneously into nude mice. Six weeks post-engraftment, we applied a treatment regimen based on the subcutaneous injection of two independent LNA antisense oligonucleotides targeting *SCAT7*, twice a week (Methods). We measured the tumor volumes after a course of four injections in two independent experiments and observed 40–50% tumor growth inhibition (TGI) in *SCAT7* LNA groups compared to the scrambled LNA control group (Fig. 8c, d; Supplementary Fig. 8c). We also monitored the weight of the mice during the tumor development and post-injections in both experiments to score for any weight loss induced by the LNA treatment. Additionally, we measured the weights and sizes of liver, spleen, and kidneys to assess the cytotoxicity of the treatment and found no difference between the two groups (Supplementary Data 3). Interestingly, the expression levels of *SCAT7* in xenografts correlated with tumor volumes (Fig. 8e, f) and tumor growth reduction was observed only in the tumors that had an efficient *SCAT7* downregulation. Then, we checked the expression levels of *FGFR3*, *FGFR4*, and *p21* (Supplementary Fig. 8d). In line with the in vitro data, *FGFR3* and *FGFR4* downregulation and *p21* upregulation was detected in tumors treated with *SCAT7* LNAs. Ki67 immunostaining, and TUNEL staining of the dissected tumors revealed the potent effect of *SCAT7* targeting on cancer malignancy in vivo (Fig. 8g). Furthermore, we implemented *SCAT7* LNA-dependent therapeutic intervention over a period of 15 days on a lung metastatic patient-derived xenograft (PDX) mouse model bearing an oncogenic mutated form of KRAS (Methods). PDX mice models injected with *SCAT7* LNA exhibited a significant reduction in the growth rate as well as the volumes (~46%) of the engrafted tumors (Fig. 8h, i; Supplementary Fig. 8e). These results collectively suggest that *SCAT7* is an oncogenic lncRNA and it can be used as a possible therapeutic target in the treatment of different cancers.

**Fig. 4** *SCAT1* and *SCAT5* act as oncogenic drivers and prognostic markers for lung-derived and kidney-derived cancers, respectively. **a** Bar graph showing the significant differential expression levels of *SCAT1* expressed as log$_2$ fold change across 10 different cancer types obtained from TCGA data sets. **b** Kaplan–Meier plots of *SCAT1* indicating overall survival of patients in LUAD (upper left panel) and LUSC (lower left panel) cancer types. The higher expression of the *SCAT1* is correlated with poor overall survival. The upper and lower right panels represent the multivariate models of LUAD and LUSC cancers, respectively, derived from Cox proportional hazard analysis and associated *p*-values were calculated using Wald statistics. **c** MTT proliferation assay of A549 (LUAD) cell line depleted with two different LNA oligos targeting *SCAT1*. **d** Cell cycle profiles of control and *SCAT1* KD A549 cells. **e** Estimation of caspase 3/7 activity in control and *SCAT1* KD A549 cells. **f** Bar graph showing the significant differential expression levels of *SCAT5* expressed as log$_2$ fold change across five different cancer types obtained from TCGA data sets. **g** Kaplan–Meier plots of *SCAT5* indicating overall survival of patients in KICH (upper left panel) and KIRP (lower left panel) kidney cancer. The higher expression of the *SCAT5* is correlated with poor overall survival. The upper and lower right panels represent the multivariate models of KICH and KIRP cancers, respectively. **h** MTT proliferation assay of Caki-2 cell line 48 h post-silencing of *SCAT5* using two different LNAs. **i** Cell cycle profiles of control and *SCAT5* KD Caki-2 cells. **j** Estimation of caspase 3/7 activity in Caki-2 cells 48 h post-silencing of *SCAT5*. The significance in figures **a** and **f** was derived using Benjamini–Hochberg's method. Note that the data presented in **c–e** and **h–j** represents the mean values of three independent experiments and statistical significance was derived using a two-tailed unpaired Student's *t*-test. Data are plotted as mean ± SD (\**p* ≤ 0.05; \*\**p* = 0.01– 0.001; \*\*\**p* < 0.001)

## Discussion

The main aim of the current investigation is based on the hypothesis that temporally expressed transcripts across S-phase may harbor crucial functions in the control of cell cycle progression and that their deregulation may underlie tumor development and progression. In the recent past, compared to protein-coding mRNAs, there is a growing appreciation for investigating the role of lncRNAs in cancer development and progression given their surprising roles in spatiotemporal gene expression. Through characterizing the dynamic transcriptional changes across S-phase in real time, we identified 1145 lncRNAs showing S-phase-specific temporal expression patterns. Enrichment of cancer-associated pathways in the GO analyses of neighboring protein-coding genes of S-phase lncRNAs, and their differential expression across Pan-Cancer data sets, indicate that S-phase lncRNAs may be strongly associated with cancer development. One of the interesting aspects of S-phase lncRNAs' differential expression analysis across TCGA data sets is that nearly 60% of them show differential expression at least in two cancer types. To confirm our hypothesis, we have selected top eight S-phase lncRNAs that show differential expression across multiple cancers and investigated their role in cancer progression via analyzing the effect of their downregulation on cancer cell hallmarks in various cancer models. Confirming our hypothesis, downregulation or

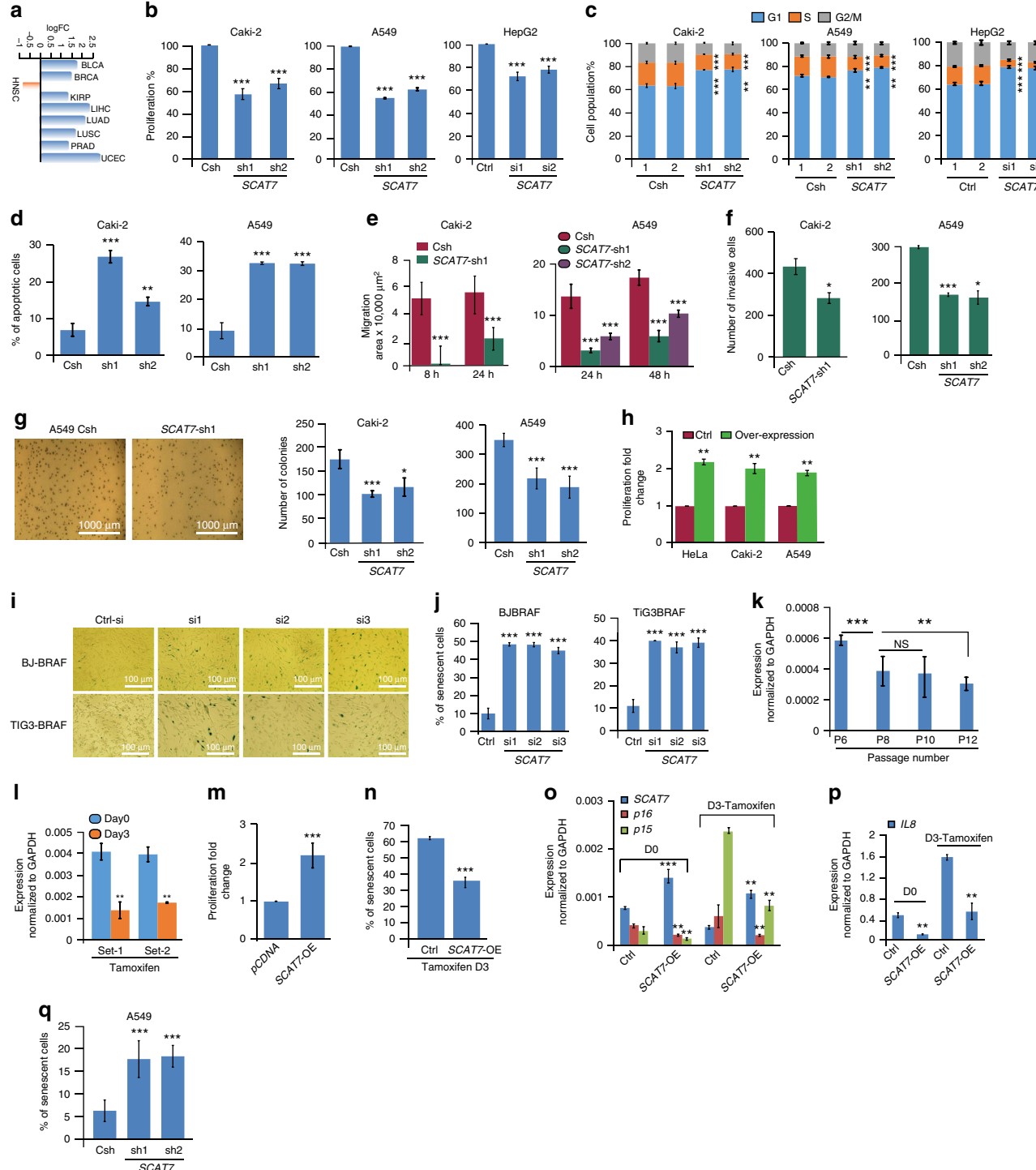

overexpression of SCATs affected cancer cell hallmarks such as cell cycle progression, cell proliferation, apoptosis, cellular senescence, and cell invasion/migration, indicating that our cell cycle-based functional screen identified potential oncogenic drivers.

DNA methylation analyses of S-phase lncRNAs revealed that DNA methylation may underlie the differential expression of S-phase lncRNAs between tumor and normal tissue pairs. For instance, SCAT3 is upregulated in six types of cancers (Supplementary Data 2) and this is consistent with its promoter's hypomethylation in several cancers (HNSC, KIRP, and LIHC). Conversely, our analysis identified a number of S-phase lncRNAs showing a hypermethylation pattern that correlates with their diminished expression in respective cancers. Therefore, our DNA methylation investigations of S-phase lncRNAs reflect the role of epigenetic alterations in modulating their transcriptional activities during tumorigenesis.

Another important aspect of the current study is the comprehensive investigation of the clinical relevance of the S-phase lncRNAs across TCGA data sets. Our analysis identified 633 S-phase lncRNAs that act as independent biomarkers with high prediction accuracy (Supplementary Fig. 2b). Strikingly, our clinical investigation indicated that KIRC, COAD, LIHC, and HNSC harbor more than 50 S-phase lncRNAs as potential independent prognostic markers. For instance, SCAT8 appeared to be the top prognostic indicator in our studies with the higher hazard ratio in our multivariate models and it also interferes with cancer cell hallmarks, indicating that it may be an oncogenic driver in multiple cancers. Further, our clinical investigations identified SCAT5 as a common independent prognostic biomarker in kidney cancers KIRP, KIRC, and KICH. Although SCAT5 was first identified in lung cancer cell patients in response to cigarette smoking[36], its role in the etiology of different kidney cancers has not been investigated. Notably, its strong prognostic significance coupled with effect of its downregulation on the cancer cell hallmarks in KIRC cell lines, indicate that SCAT5 could be a potential therapeutic target in the treatment of kidney cancers. Similarly, SCAT1, which is upregulated in 10 different cancers, shows a functional involvement and independent prognostic capacity in four cancers including LUAD and LUSC. Thus, our functional and clinical investigations on SCAT8, SCAT5, and SCAT1, suggests that our cell cycle-based functional screen indeed has identified potential lncRNA-based cancer drivers. Importantly, our analysis indicates that clinically relevant S-phase lncRNAs can be considered as independent prognostic biomarkers with a strong potential to be used in the clinical setting for better risk stratification and prediction of clinical outcome.

For a further understanding of the mode of action of S-phase-enriched lncRNAs in tumorigenesis, we performed a detailed mechanistic investigation using SCAT7 as a model. The consistency of the obtained phenotypic effects upon SCAT7 knockdown in different model systems indicates a conserved functional interaction between SCAT7 and its downstream target genes to ensure cellular homeostasis. The crosstalk between cell cycle progression, apoptosis, and cellular senescence has been firmly established in the maintenance of cellular homeostasis[37] and our data clearly demonstrated that the silencing of SCAT7 strongly interferes with these interconnected biological processes, thereby affecting cellular homeostasis. Upon SCAT7 knockdown different types of proliferative cell lines exhibit the characteristic features of cellular senescence including β-galactosidase secretion, cell cycle arrest, and induction of different tumor suppressor genes[38] (p21, p16, and p15). This is consistent with the enrichment of biological processes such as cell cycle, apoptosis, cell proliferation, and cell senescence. Moreover, SCAT7 depletion in multiple cell lines show accumulation of cells in G1 phase and S-phase progression defects. The drastic reduction in the EdU+ve cells upon SCAT7 depletion is also a clear indication of the diminished S-phase within DNA-replicating cells. Reduced G1 to S phase progression in SCAT7 depleted cells may be due to CCND1 downregulation and RB hypophosphorylation. CCND1 downregulation could be the result of inhibitory effects of FOXO1/3 factors due to the reduced AKT activity[39,40]. Though these observations support the role of SCAT7 in G1 to S phase progression, more work is needed to ascertain the functions of SCAT7 in S-phase progression. Thus, our functional investigations unequivocally implicate SCAT7 in the cellular homeostasis through regulating the cancer cell hallmarks (Fig. 8j).

The dissection of the regulatory pathways mediated by the action of SCAT7 indicated its crucial involvement in regulating pivotal signaling pathways across multiple cancer models. lncRNAs have been reported to regulate signaling pathways like Wnt[41,42], Notch[43], PI3K/AKT[44], and RAS/MAPK[45]. However, the regulation of FGF/FGFR signaling by lncRNAs has not been explored mechanistically in great detail. Given that several FGF/FGFR members were deregulated upon SCAT7 knockdown in multiple cancer models, we hypothesized a genuine connection between SCAT7 and FGF signaling in the context of cancer. The functional conservation of SCAT7-hnRNPK-YBX1 RNP complex in different cancer models presents a mechanistic model of FGF/FGFR regulation. The recruitment of the SCAT7-hnRNPK-YBX1 RNP complex at the promoter regions of FGFR2 and FGFR3 promotes transcriptional activation of the FGF/FGFR pathway, leading to sustained cell proliferation via PI3K/AKT and Ras/MAPK pathways (Fig. 8k). It is pertinent to note that the identification of hnRNPK and YBX1 as the interacting proteins of SCAT7 fits well in the current understanding of the functional roles of hnRNPK and YBX1 in the regulation of cell cycle

**Fig. 5** SCAT7 acts as an oncogenic driver in renal, lung, and liver cancers. **a** SCAT7 expression as $\log_2$ fold change across cancer types from TCGA data sets. The significance was obtained using Benjamini–Hochberg's method. **b** MTT of Caki-2, A549, and HepG2 cells upon SCAT7 KD with two shRNAs or siRNAs. **c** Cell cycle profiles of Caki-2, A549, and HepG2 cells after shRNA or siRNA-based SCAT7 KD. **d** Percentage of apoptotic cells in SCAT7 KD Caki-2 and A549 48 h post-seeding. **e** Migration areas for stable SCAT7 KD Caki-2 and A549 cells were calculated with respect to a starting ($t = 0$) migration control area for each cell line. **f** Matrigel transwell assay in Caki-2 and A549 SCAT7 stable KD cells. The number of invasive cells was counted 24 h post-seeding. **g** Soft agar colony-forming assay using Caki-2 and A549 KD cells. **h** MTT of HeLa, Caki-2, and A549 cells overexpressing SCAT7. **i** Colorimetric β-galactosidase staining of BJ-BRAF and TIG3-BRAF human fibroblasts 72 h post-silencing SCAT7 using three siRNAs. Senescent cells are in dark blue color. **j** Quantification of senescent cells upon SCAT7 KD in BJ-BRAF and TIG3-BRAF cells shown as percentage of the whole cells populations. **k** SCAT7 qPCR in serial passages of BJ-BRAF cells. NS, not significant. **l** SCAT7 expression in BJ-BRAF cells at day 0 and day 3 upon tamoxifen-induced senescence (200 nM) at one passage interval. **m** MTT assay of BJ-BRAF cells overexpressing SCAT7 compared to empty vector. **n** Percentage of positively stained senescent cells 3 days post-tamoxifen treatment in control and SCAT7-overexpressing BJ-BRAF cells. **o, p** Expression of SCAT7, p16, p15 (**o**) and IL8 (**p**) in control BJ-BRAF and SCAT7-overexpressing cells at day 0 and day 3 post-treatment with tamoxifen. **q** Quantification of senescent cells upon SCAT7 KD in A549 cells. The values are expressed as percentage of the whole cell population. Note that the data presented in **b**–**h** and **j**–**q** represents mean values of three independent experiments and the statistical significance was derived using a two-tailed unpaired Student's t-test. Data are plotted as mean ± SD (*$p \leq 0.05$; **$p = 0.01$–0.001; ***$p < 0.001$)

progression, gene expression, and cell signaling[34,35,46]. However, it will be interesting to scrutinize whether cell identity, as shown in the KIRC model, affects *SCAT7*/RNPs interactions in a lineage-specific manner. Nevertheless, lineage-specific interactions between lncRNAs and RNPs have been shown to be involved in developmentally regulated biological functions[23,47–49].

The functional significance of FGF/FGFR members in normal development, maintenance of stem cell properties, and senescence is fairly well understood[50]. Regulation of *FGF/FGFR* members by

*SCAT7* in HeLa, KIRC, and LUAD cancer models highlight the functional role of *FGF* signaling in *SCAT7*-mediated tumor initiation and progression. Consistent with our data, several investigations have implicated FGF signaling in renal and lung cancers. For instance, a SNP in *FGFR2* was shown to affect the progression-free survival alone in the metastatic KIRC patients undergoing anti-VEGF-targeted therapy[51]. Also, a missense mutation in *FGFR2* was found to drive a durable response to nucleolin-targeted therapy in metastatic KIRC[52]. Despite the fact

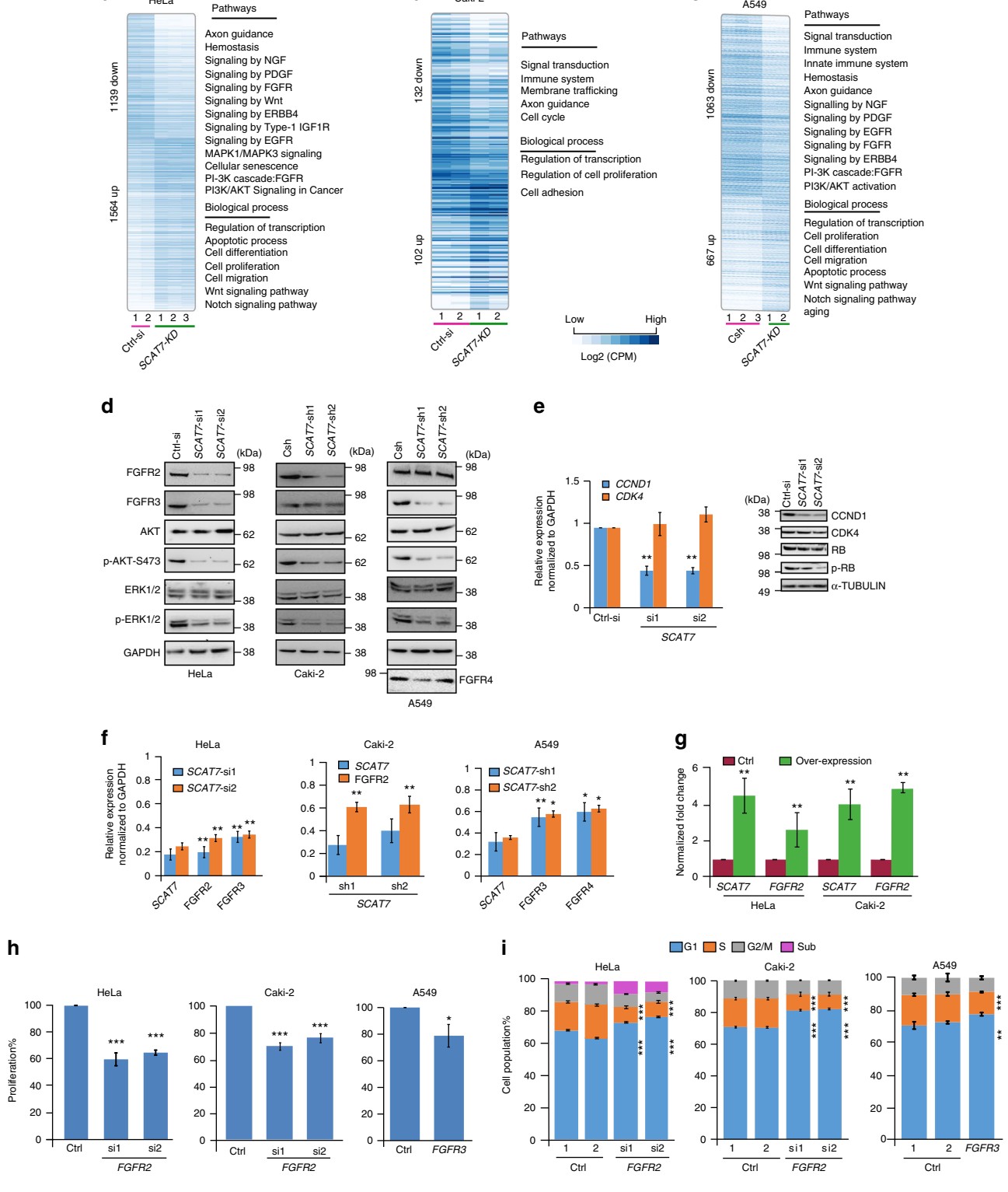

that various FGFR members harbor activating mutations in lung cancer patients and confer acquired resistance to tyrosine kinase inhibitors (TKIs)[53,54], only a handful of studies have addressed the role of these mutations in LUAD tumorogenesis. Tchaicha et al. generated the first genetically engineered lung cancer mouse model harboring an FGFR mutation in p53 null background[55]. The engineered mouse model showed more than 50% tumor regression when treated with a pan-FGFR inhibitor. More recently, Manchado et al. reported a combinatorial approach including FGFR1 inhibitor to overcome the adaptive resistance to MEK inhibitor in KRAS-mutant LUAD[56]. Considering the functional nexus between *SCAT7* and FGF signaling, targeting *SCAT7* alone is sufficient to inhibit tumor progression via repressing different members of the FGF/FGFR pathway. Supporting this, *SCAT7* LNA-based treatment used in this study established a possible therapeutic intervention regimen for multiple cancers in vivo. Based on these observations, we suggest that a combinatorial treatment strategy involving *SCAT7* repression alongside treatment with potent FGFRs inhibitors or TKIs will hold promise for lncRNA-based therapeutics.

Collectively, we provide a comprehensive list of lncRNA-based oncogenic drivers with potential prognostic value. More importantly, this systematically analyzed functional and clinically relevant lncRNAs can serve as a resource for delineating the functional link between lncRNA and tumor development and progression.

## Methods

**Cell lines and cell synchronization**. The adherent Caki-2, 786-O, and HepG2 cell lines were purchased from CLS-GmbH (Germany). The LUAD cell lines A549 and H2228 were kindly provided by Bengt Hallberg's lab at the department of Medical Biochemistry and Cell Biology, University of Gothenburg. HeLa and MCF-7 cell lines were routinely maintained in our lab. The immortalized human fibroblasts cell lines, BJ-BRAF and TiG3-BRAF, were kindly provided by Andres Lund's lab (Biotech Research and Innovation Center, University of Copenhagen). We cultured Caki-2, 786-O, and H2228 cell lines in RPMI-1640 medium (Gibco, Life Technologies, USA). A549, BJ-BRAF, TiG3-BRAF, and MCF-7 cell lines were maintained in DMEM medium (Gibco, Life Technologies; USA). HeLa and HepG2 cell lines were maintained in MEM medium (Gibco, Life Technologies; USA). All media were supplemented with 2 mM L-glutamine, 1× penicillin-streptomycin antibiotic, and 10% fetal bovine serum. All cell lines were tested negative for Mycoplasma contamination. For RT-qPCR validation, HeLa cells were synchronized following the addition of 2 mM of thymidine into a fresh medium for 10 h then aspirating the medium and adding hydroxyurea to a final concentration of 1 mM overnight. The synchronized cells were collected at 2, 3.5, 5, and 9 h representing T1, T2, T3, and T4, respectively. For serum starvation, HeLa, A549, and Caki-2 cells were cultured in serum-free media for 36–44 h. The synchronized HeLa cells were collected at the indicated time points while Caki-2 cells were harvested at 4, 8, and 20 h representing T1, T2 and T3, respectively. In A549 cells, T1, T2, and T3 synchronized cells were harvested at 4, 6, and 18 h, respectively.

**Nascent RNA capture assay**. To capture nascent RNA and total RNA from different stages of S-Phase, on day 1 HeLa cells were plated at 500,000 cells per F75 plate, to ensure cell confluency of around 35% during synchronization. On day 2 cells were incubated for 10 h in medium supplemented with 2 mM thymidine (Sigma). After 10 h, cells were washed with PBS and incubated with medium supplemented with 1 mM hydroxyurea (Sigma) for 14 h. The synchronized cells are now just at the beginning of S-phase with a confluency of 70%, which allowing them to resume dividing upon release from cell cycle block without suffering from

contact inhibition. After a brief PBS wash, medium is then changed back to normal, and the cells are allowed to progress through S-phase in a synchronized manner. The time at which the block is released is thereafter termed "T0". The labeling of nascent RNA was carried out for 2-h periods at different stages of S-phase. At the beginning of the labeling period, media was supplemented with EtU (Invitrogen) to a final concentration of 1 mM, and cells were harvested 2 h later. The labeling periods were defined as follows: T0h–T2h (beginning of S-phase), T1.5h–T3.5 h (middle of S-phase), and T3h–T5h (end of S-phase). This labeling protocol ensured that, for all three S-phase samples, the labeled nascent transcripts would represent only 2 h of transcription, in order to provide a detailed and accurate picture of the timing of transcriptional events occurring throughout S-phase. For unlabeled total RNA samples, we followed the same procedure but with medium without EtU supplement.

Cells from EtU labeled and unlabeled samples were harvested at the end of their respective 2 h labeling period, and RNA were extracted with Tri reagent (Ambion). We performed DNA digestion with RQ1 DNase I (Promega) for 1 h at 37 °C, and re-extracted RNA with Tri reagent. An aliquot of 10 µg RNA was re-suspended in low volume and rRNA depletion was carried out with Ribominus kit (Invitrogen). From each S-phase sample, 2, 3 µg of rRNA-depleted RNA was biotinylated with Click-It Nascent RNA Capture kit (Invitrogen) following manufacturer's protocol. The biotinylated RNA was captured using streptavidin magnetic beads. The captured biotinylated RNA was eluted by incubating the beads in 200 µl buffer containing 2 mM biotin, 1 M NaCl, 50 mM MOPS, 5 mM EDTA, 2 M 2β-Mercaptoethanol, pH 7.4 for 3 min at 95 °C. Supernatant containing eluted RNA was recovered immediately after heating and the RNA was precipitated in 30 µl 3 M sodium acetate (pH 5.2), 1 µl glycoblue (Invitrogen), 750 µl 100% ethanol. RNA was then resuspended in nuclease-free water, dosed on a Qubit fluorometer (Invitrogen), and RNA size profile was assessed on Bioanalyzer (Agilent) with RNA pico 6000 kit and sent for library construction and sequencing on a Solid platform at Uppsala Genome Center. For unlabeled RNA samples, total RNA was extracted and subjected to rRNA depletion, and used for RNA-sequencing.

**Knockdown of target genes and cloning**. We performed transient transfection using siRNA, esiRNA, and antisense LNA ¨GapmeR molecules. Scramble siRNA, custom-made siRNAs, and pre-designed esiRNAs were designed and synthesized by Sigma-Aldrich. We obtained both in vivo and in vitro grade negative control and custom-made LNA ¨GapmeR molecules from Exiqon. Transfection was carried out in standard 24-well plates using Lipofectamin® RNAiMAX transfection reagent (Invitrogen, California) according to the manufacturer's instructions with a final concentration of 35 pmol and 20 pmol/well of siRNA and LNA, respectively. The transfections were performed in three biological replicates and the efficiency of KD was confirmed by RT-qPCR. We generated stable cell lines using Lentifect™ Purified shRNA lentivirus particles designed and synthesized by GeneCopoeia™ to target *SCAT7* or scramble negative control. The transduction efficiency was visualized by the integrated GFP reporter gene. The stable 786-O, Caki-2, and A549 KD cells were selected using 2 µg/ml, 3 µg/ml, and 2.5 µg/ml of puromycin, respectively. All custom-made siRNAs, LNAs, and shRNAs particles were designed to target the unique non-overlapping exons of the respective transcript. For *SCAT7* overexpression, the *SCAT7* fragment was cloned into pGEM-T Easy vector and the correct orientation was verified by sequencing. The confirmed clone was digested using NotI and HindIII enzymes and sub-cloned into the mammalian overexpression vector pcDNA3.1. HeLa, Caki-2, and A549 cells were transfected with either 1 µg of pcDNA3.1-*SCAT7* vector or pcDNA3.1 empty vector using Lipofectamin® 2000 transfection reagent. The RNA levels were measured with RT-qPCR. The sequences of siRNAs, shRNAs, LNAs, primers, and antibodies are listed in Supplementary Data 4.

**Flow cytometry, cell cycle, and Click-iT EdU assay**. We assessed the cell cycle profiles of transient KD cells and control cells 48 h post-transfection. The media were aspirated, and the cells were washed with PBS, trypsinized, pelleted by centrifugation, washed twice with cold PBS, fixed with ice-cold 70% ethanol, and stored at −20 °C for at least 2 h. The fixed cells were re-collected by centrifugation, re-suspended in PBS, and kept for 30 min at 37 °C. Then, cells were collected and stained with PI solution containing 1% RNase A in PBS and kept at 4 °C for at least 4 h. The PI-stained cells were assayed using Eclipse single-cell flow cytometry

**Fig. 6** *SCAT7* regulates cell cycle progression and cell proliferation via FGF signaling. **a–c** Heatmaps showing upregulated and downregulated genes with corresponding molecular pathways and biological processes upon silencing of *SCAT7* in HeLa (**a**), Caki-2 (**b**), and A549 (**c**) cell lines. **d** Western blot showing the proteins levels of FGFR2, FGFR3, AKT, Ser 473 Phospho-AKT (p-AKT S473), ERK1/2, and Phoshpo-ERK1/2 (p-ERK 1/2) upon silencing of *SCAT7* in HeLa, Caki-2, and A549 cell lines. FGFR4 protein levels were only investigated in A549 cells. **e** Real-time qPCR validation (left panel) and Western blot (right panel) showing a significant reduction in the expression levels of *CCND1* but not *CDK4*. **f** Real-time qPCR validation of the expression levels of *SCAT7* and its targets *FGFR2*, *FGFR3*, and *FGFR4* in HeLa, Caki-2, and A549 cells upon *SCAT7* KD with two independent siRNAs or shRNAs. **g** Expression of *SCAT7* and its target *FGFR2* in HeLa and Caki-2 cells overexpressing *SCAT7*. Data are shown as relative fold change normalized to GAPDH. **h** MTT assay in HeLa and Caki-2 cells, after transfection with siRNAs targeting *FGFR2*, and A549 cells, transfected with esiRNA to silence *FGFR3*. **i** Cell cycle profiles of HeLa and Caki-2 cells transfected with siRNAs targeting *FGFR2* (left and middle). The right panel shows the cell cycle profile of A549 cells transfected with *FGFR3* esiRNA. Note that the data presented in **e–i** represent mean values of three independent experiments and the statistical significance was derived using a two-tailed unpaired Student's *t*-test. Data are plotted as mean ± SD (*$p \le 0.05$; **$p = 0.01$–$0.001$; ***$p < 0.001$)

system ec800 and data were analyzed with the manufacturer's software. The cell cycle profiling was validated on another system using NucleoCounter NC-3000 platform (Chemometec, Denmark). The fixed cells were stained with DAPI solution provided by the manufacturer and analyzed according to manufacturer's instructions. All KD experiments were assayed three times independently and statistical significance was derived using two-sided unpaired Student's t-test. The EdU incorporation assay was performed according to the manufacturer's protocol

using the Click-iT™ EdU Alexa Fluor™ 488 Imaging Kit and the green fluorescence wad detected using the EVOS FL Auto Cell Imaging System.

**Northern blotting analysis.** Total RNA was extracted from wild-type A549 and SCAT7-KD cells and an equal amount of 10 μg of each sample was loaded into 1% formaldehyde agarose gel alongside 1.0 μg of 0.5–10 kb RNA ladder (Invitrogen). We used 1 × MOPS-formaldehyde as a running buffer in RNase-free conditions.

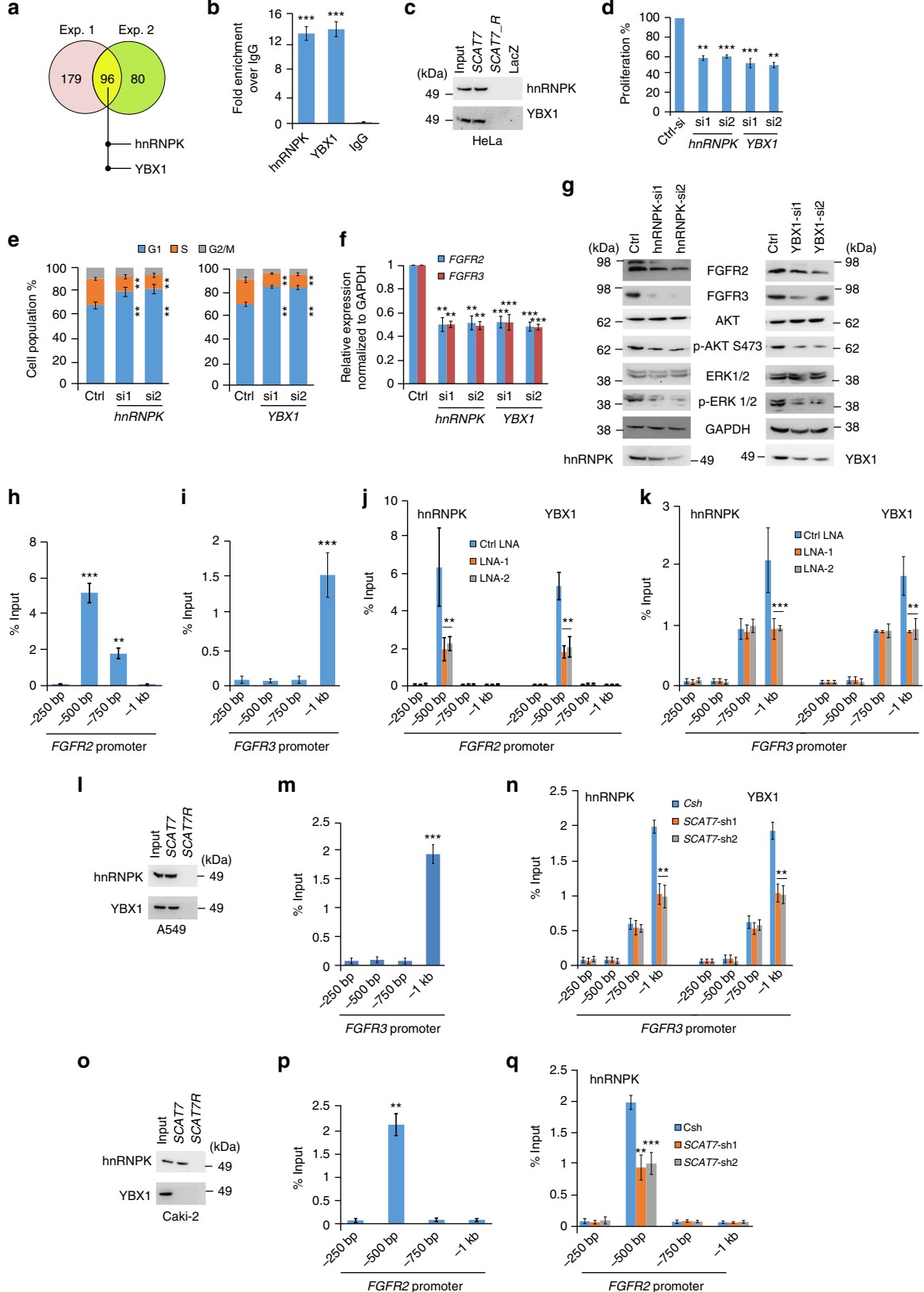

Following the gel electrophoresis, the wet transfer was performed overnight using $20 \times$ SSC as a transfer buffer at room temperature. After the transfer, the membrane was fixed under UV then it was cut and immediately incubated 3 h with rapid-hyb hybridation buffer (GE Healthcare Life Sciences) for pre-hybridization at 42 °C. The radioactive probes were prepared freshly for *SCAT7* and RNA ladder. For *SCAT7*, we utilized the T4 PNK enzyme (NEB) to label the *SCAT7* short probes (the same probes used in ChOP experiment) using $\gamma^{32-P}$ ATP (PerkinElmer). For the RNA ladder probes, we used the RNA ladder as a template to synthesis the complementary probes using labeled α-dCTP in a reverse transcription reaction. The probes were purified using G25 columns (GE Healthcare Life Sciences) then they were hybridized separately with the corresponding membrane at 42 °C overnight. The membrane was washed twice at room temperature with low stringent buffer ($2 \times$ SSC and 0.1% SDS) for 10 min each followed by two washes of warm high stringency buffer (0.1 × SSC and 0.1% SDS) at 55 °C for 15 min each. The membranes were left for exposure overnight and scanned with Fuji FLA7000 phosphoimager.

**Overview of annotations and tools used in this study.** The hg19 (GRCh37) genome version for alignment and transcript annotation from GENCODE version 19 (equivalent Ensembl GRCh37)[57] was used for processing RNA-sequencing samples. Color space reads from SOLID sequencing platform were aligned using LifeScope, and HISAT[58] aligner was used for reads from Illumina sequencing platform. The reads for gencode transcripts were quantified using HTSeq-count[59] with 'intersection-strict' mode.

**Sequencing and alignment.** For each S-phase time point, RNA from two independent experiments were pooled into one sample and deep sequenced on SOLID sequencing platform at Uppsala Genome Center. The alignment resulted in 18.5, 39.3, 21.3, and 29 million mappable 75 bp reads for unsynchronized, S-phase T1, S-phase T2, and S-phase T3 EtU-labeled samples, respectively. For unlabeled samples, we obtained 19.6, 30, 29, and 27 million mappable 75 bp reads for unsynchronized, S-phase T1, S-phase T2, and S-phase T3 samples, respectively.

**Transcript annotation and read quantification.** The uniquely mapped reads were assigned to long noncoding transcripts from Gencode v19 annotation and obtained 4039 and 3966 lncRNAs from EtU labeled and unlabeled samples, respectively, which are expressed at least at one time point of S-phase. The obtained reads were normalized to their library sizes and transcript length (reads per kilobase of transcript per million—RPKM normalization). The log-fold change values were derived from obtained RPKM values by comparing individual time-point against unsynchronized samples.

**Time-series analysis to identify lncRNAs across S-phase.** The lncRNAs having log-fold change greater than one in at least one S-phase time-points over unsynchronized sample were taken for further analysis. We obtained 1734 and 1674 lncRNAs in EtU labeled and unlabeled samples respectively. The Short Time-series Expression Miner (STEM) clustering which is specifically designed to handle short time-series gene expression data was used to find the significant temporal patterns in S-phase-derived lncRNAs. The significant temporal expression patterns were obtained from expression profiles of S-phase lncRNAs in three different time-points (2, 3.5, and 5 h) by assuming unsynchronized sample as a base time point. There were 1145 (out of 1734) lncRNAs in EtU labeled and 937 (out of 1674) lncRNAs in unlabeled samples, showed significant temporal expression patterns across S-phase as in Fig. 1c and Supplementary Fig. 1a.

**E2F1-bound EtU-labeled S-phase lncRNAs.** E2F1 transcription factor ChIP-seq peaks were obtained for HeLa cell line from ENCODE (ENCSR000EVJ) hg19 processed files. Using Homer 'annotatePeaks.pl', the E2F1 peaks were associated with promoter of the transcripts from Gencode v19 annotation. Individual peaks are assigned to the promoter if it is within ±2 kb to the TSS of a transcript.

**Functional significance of nearby protein-coding genes.** The nearby protein-coding genes to EtU labeled, unlabeled and common S-phase lncRNAs were tested for their functional significance. We observed most of the first hit to the S-phase lncRNAs are within 50 kb distance. Protein-coding gene first hit within 50 kb window relative to S-phase lncRNAs from EtU labeled and unlabeled samples were extracted using BedTools 'closest' with parameters '-D ref' and further used for functional enrichment analysis. GeneSCF v1.1[24] was used to perform gene enrichment analysis. It is important for an enrichment tool to be updated frequently to get more reliable results and avoid misinterpretation of the data[60]. The major advantage of GeneSCF is that it uses the updated or recent terms and updated genes from the databases. GeneSCF is more reliable compared to most enrichment analysis tools because it has real-time based enrichment analysis feature. The code for GeneSCF is also regularly updated to accommodate the changes from the source database like KEGG, Gene Ontology, and Reactome. In this study, the terms are considered significant only if it is enriched with $p < 0.05$ and with minimum of 10 genes involved in a particular function. These genes were tested against two different functional annotations, Gene Ontology biological process (GO-BP), and KEGG pathways by assuming all protein-coding gene numbers from Gencode v19 as background (20,345) or reference set.

**Processing S-phase lncRNAs in multiple cancers from TCGA.** We have aligned RNA-sequencing reads using HISAT from 16 cancer types from TCGA. The read abundance was quantified for Gencode v19 transcripts using HTSeq-count in "intersection-strict" mode (options --no-mixed --no-discordant --no-unal –known-splicesite-infile) for obtaining the read counts per transcript. The obtained reads (counts) were normalized to their library sizes and transcript length (RPKM normalization). Using these normalized counts, the significance of differential expression between normal samples and corresponding solid tumor was obtained using Wilcoxon signed-rank non-parametric test and corrected for multiple testing with Benjamini–Hochberg's method (FDR). The cell cycle-associated lncRNA to be considered as significantly differentially expressed cancer-associated transcript, we used log-fold change ± 2 and FDR < 1E–004 in at least one cancer type as criteria. These cancer-associated lncRNAs were further subjected to clinical analysis. The information on each cancer type and the number of tumor and normal tissue samples processed for individual cancer type is provided in Supplementary Data 2.

**Processing S-phase lncRNAs for TCGA CpG methylation data.** Processed differential methylation data was downloaded from COSMIC[28] repository for GRCh37 genome and version 74 transcript annotation equivalent to Gencode v19 annotation. COSMIC analysis predicted differentially methylated regions (DMRs) by comparing the beta-values derived from TCGA data level 3 for tumor and matched normal samples. The cancer types containing more than 19 normal samples were considered for statistical analysis. The $p$-values for the comparisons were obtained using Mann–Whitney test and corrected $p$-values using family wise error rate (FWER). The significant DMRs were filtered on the basis of FWER < 0.05. The above information for statistical analysis was extracted from COSMIC repository.

The significant DMRs from COSMIC were assigned to promoter (−2000 bp and +500 bp from TSS) and gene body (TSS + 550 to length of transcript) regions of the cancer-associated lncRNAs from our study. We considered any lncRNA as differentially methylated only if more than 10 patient samples support the methylation status and also absence of any patients which supports opposite methylation pattern for the same region. As an example, the promoter region has to be considered as hypermethylated if there were at least 10 patients supporting hypermethylation status but no patients (null) show hypomethylation at the same promoter region or other regions within the transcript. All presented hypo and

**Fig. 7** *SCAT7* interacts with hnRNPK and YBX1 to regulate cell proliferation and cell cycle progression. **a** Venn diagram showing *SCAT7* interacting proteins in HeLa cells identified using ChOP-MS in two independent biological replicates. **b** RIP using hnRNPK or YBX1 antibody followed by qPCR for *SCAT7*. **c** Validation of *SCAT7* interaction with hnRNPK and YBX1 by ChOP followed by immunoblotting in HeLa cells. LacZ and *SCAT7* reverse biotinylated probes were used as negative controls. **d** MTT in HeLa cells after transfection with two siRNAs targeting hnRNPK or YBX1. **e** Cell cycle analysis upon hnRNPK and YBX1 KD in HeLa cells. **f** *FGFR2* expression by real-time qPCR in hnRNPK and YBX1 KD HeLa cells. **g** Western blot of FGFR2, FGFR3, AKT, Ser 473 Phospho-AKT (p-AKT S473), ERK1/2, and Phoshpo-ERK1/2 (p-ERK 1/2) in hnRNPK and YBX1 KD HeLa cells. **h, i** ChOP followed by qPCR for *SCAT7* enrichment at *FGFR2* (**h**) and *FGFR3* (**i**) promoters in HeLa cells. Four primer pairs were used to assess the occupancy at every 250 bp upstream of *FGFR2* and *FGFR3* TSS. **j, k** ChIP using hnRNPK or YBX1 antibody followed by qPCR for hnRNPK and YBX1 occupancy at *FGFR2* (**j**) and FGFR3 (**k**) promoters in control and *SCAT7* KD HeLa cells. **l** Interaction of *SCAT7* with hnRNPK and YBX1 by *SCAT7* ChOP followed by immunoblotting in A549 cells. **m** ChOP followed by qPCR for *SCAT7* enrichment at *FGFR3* promoter in A549 cells. **n** hnRNPK or YBX1 ChIP followed by qPCR depicting the occupancy of hnRNPK and YBX1 at the *FGFR3* promoter in control and *SCAT7* KD A549 cells. **o** *SCAT7* ChOP followed by immunoblotting with hnRNPK or YBX1 antibody in Caki-2 cells. **p** ChOP followed by qPCR quantification of *SCAT7* enrichment at *FGFR2* promoter in Caki-2 cells. **q** hnRNPK ChIP followed by qPCR depicting the occupancy of hnRNPK at *FGFR2* promoter in control and *SCAT7* KD Caki-2 cells. Note that the data presented in **b**, **d–f**, **h–k**, **m**, **n**, **p**, and **q** represent mean values of three independent experiments and the statistical significance was derived using a two-tailed unpaired Student's *t*-test. Data are plotted as mean ± SD (*$p \leq$ 0.05; **$p = 0.01–$ 0.001; ***$p < 0.001$)

hypermethylation lncRNAs has FWER < 0.05 obtained with matched normal and tumor comparison in respective cancer types. The number of tumor samples used for the methylation analysis for each cancer type is included in Supplementary Data 2 and the detailed description on COSMIC differential methylation analysis is available at "http://cancer.sanger.ac.uk/cosmic/analyses" under "Methylation" sub-heading.

**Clinical investigation of S-phase lncRNAs in TCGA tumors.** For clinical investigations, we have chosen 1145 lncRNAs that were showing significant temporal expression in EtU-labeled samples. The selected lncRNAs were dichotomized with respect to the expression cut-off based on either mean, median, or quartiles and considered whichever gave the best discrimination. Thereafter, overall survival rate in patients above and below the selected cut-off levels, was calculated according to the KM method, and the log-rank test was used to assess differences in survival. We used survival ROC R package, in which the time-dependent ROC is calculated using the KM estimator for the expression cut-offs[61]. Points above the diagonal represent good classification results (better than random), points below the line poor results (worse than random). AUC < 0.5 was filtered to prevent randomness in the discrimination of poor and good survival groups. To further investigate the relation to survival, dichotomization and continuous levels of S-phase lncRNAs were assessed in a Cox regression model and hazard ratios calculated. Based on expression distribution, Student's t-test or Wilcoxon rank sum test was used to evaluate mean expression of SCATs in relation to clinical parameters such as node (N0 vs. N1_N3), stage (Tumor stage 1 & 2 vs. Tumor stage 3 & 4), size (Tumor size 1 & 2 vs. Tumor size 3 & 4), metastasis (M0 vs. M1), grade (Tumor grade 1 & 2 vs. Tumor grade 3 & 4), and age ( < median age vs. > median age). We constructed a multivariate Cox regression model including clinical parameters and S-phase lncRNAs expression that were significant in univariate analysis to determine the prognostic effect.

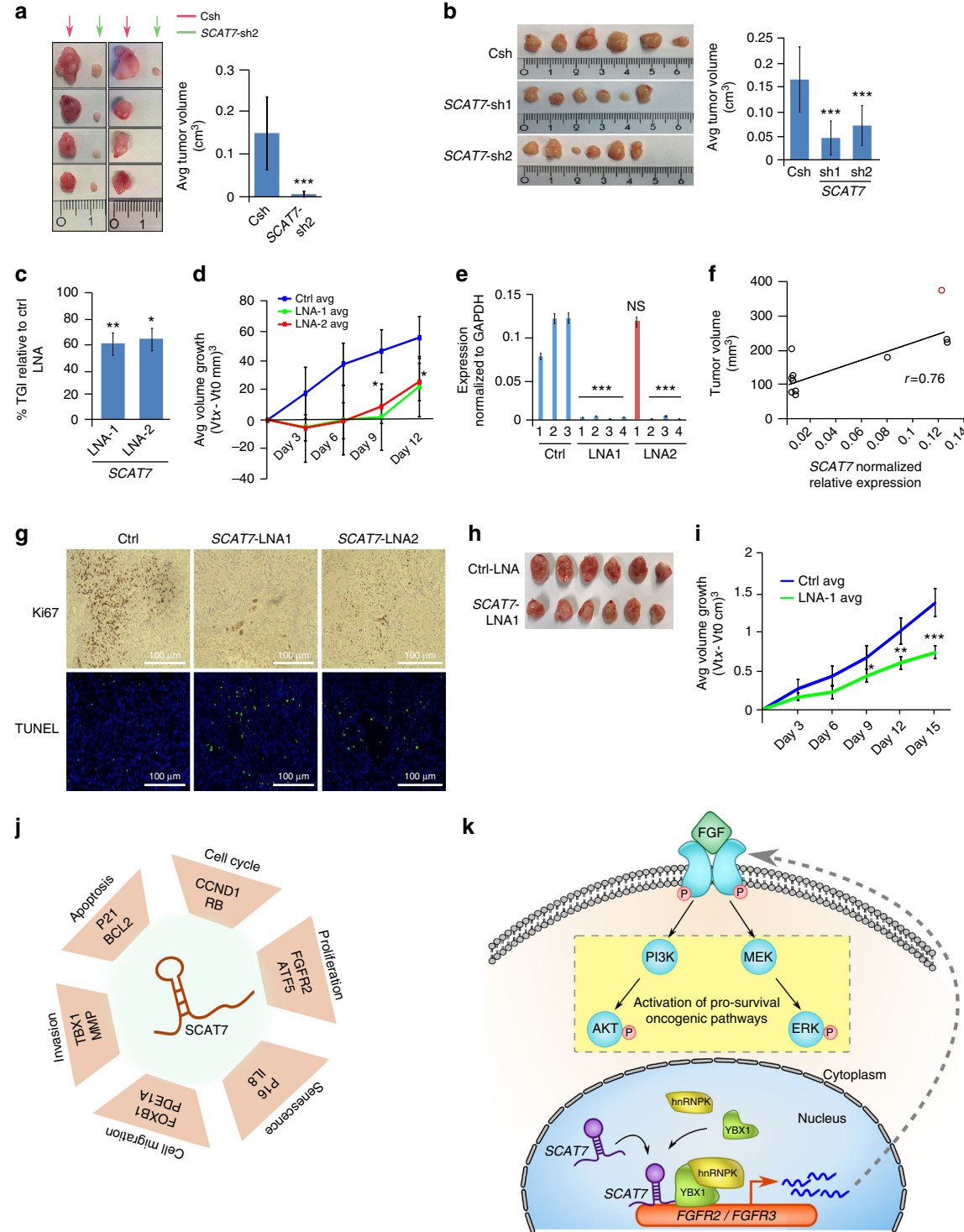

Brier score was used to assess the prediction error in the model. Pec from the R package was used for Brier score prediction. All the statistical analyses were performed using R package and $p < 0.05$ was considered as significant. We considered the lncRNA as a potential prognostic marker, if they showed significant survival difference in log-rank test and had a statistically significant hazard value in coxph regression analysis either in dichotomization or in continuous scale. Survival analysis and coxph regression analyses were performed using Survival package in R.

**Sequencing and differential expression analysis**. The reads were cleaned for adapter sequence using Trimmomatic (v 0.32) and the alignment of cleaned reads was done with hg19 reference genome using HISAT aligner (mode: --sensitive --qc-filter). We used Gencode v19 annotation to quantify the read abundance using HTSeq-count. The differential gene expression between knockdown (KD) and control cells was analyzed using bioconductor package edgeR[62]. Genes were tested for differential expression only if expression (CPM) is greater than 1 in at least two samples of comparison groups. The significant candidates were filtered with log-fold change $> \pm 1$ and FDR $\leq 0.05$.

**Pathway enrichment analysis of deregulated genes**. The significantly deregulated genes from siRNA knockdown samples were tested for pathway enrichment using tool GeneSCF v1.1. The parameters for GeneSCF were set to database Reactome with background genes 20,345 (all protein-coding genes from Gencode v19 annotation used in this study). The enriched functions were filtered based on $p < 0.05$ and minimum of 10 genes involved in corresponding process.

**Proliferation and vitality assays**. We assayed the proliferation capacity of transiently transfected cells 48 h post-transfection using CellTiter 96® Non-Radioactive Proliferation Assay kit (Promega, USA) with some modifications to the manufacturer's protocol. The media were aspirated and cells were washed once with PBS, and 425 μl of fresh medium plus 75 μl of MTT dye were added and incubated at 37 °C in dark for 4 h. Then, each reaction was terminated using 500 μl of stop solution and the cells were kept overnight in dark at 4 °C to solubilize the MTT dye. The dye intensity was measured using microplate reader at 570 nm. Standard deviation (SD) and statistical significance were derived from three independent experiments. For assaying the proliferation capacity of stable KD cell lines, we seeded the same number of control and KD cells and applied the same protocol used in the transient transfection experiments. We performed the vitality assay for transient KD cells and stable KD cell lines using NucleoCounter NC-3000 platform. The cells were harvested and stained with a mixture of VB-48™, PI, and acridine orange dyes according to the manufacturer's instructions. The results were viewed and analyzed by the manufacturer's software.

**Apoptosis and senescence assays**. We performed apoptosis assay for transiently transfected cells 48 h post-silencing using Caspase-Glo® 3/7 Assay (Promega, USA) and measured the luminescence according to the manufacturer's instructions. For stable KD cells, in addition to the Caspase-Glo® 3/7 Assay, we performed fluorescent-based Guava Caspase 3/7 FAM Assay (Merk Millipore, Germany) following the manufacturer's instructions. We analyzed the caspase 3/7 FAM activity using NucleoCounter NC-3000 platform. For detection of senescence in fibroblasts and HeLa cells, we carried out transient transfections for 72 h. In case of A549 stable KD cells, we seeded the cells at less confluency for 72 h. We then detected the senescent cells using Senescence β-Galactosidase Staining Kit (Cell Signaling Technology, USA) following the manufacturer's instructions.

**Soft agar colony-forming assay**. We used a standard procedure for soft agar colony-forming assays in a 24-well plate. In each well, we plated 500 μl of 1:1 mix of 1% molecular biology grade agar and 2× medium (RPMI 1640 or DMEM) supplemented with 20% FBS, then agar layer was left to harden for 30 min. The soft agar layers were prepared by mixing either stable KD cells or control cells with a 500 μl mix of 1:1 0.6% agar and 2× medium supplemented with 20% FBS. The soft

agar layers containing 2500 cells/well were left for 15 min to harden and then we added 500 μl of 1× RPMI 1640 or DMEM supplemented with 10% FBS on the top of the agar layers to prevent any possible dehydration. For each cell line, we plated 10 wells. After 10 days of incubation at 37 °C, we captured pictures of each well using an automated Z-stack function of the EVOS™ FL Auto Cell Imaging System (ThermoFisher Scientific). We counted the number of total colonies in each condition and measured the surface area of at least 20 representative colonies of each cell line. Statistical significance was derived using two-sided unpaired Student's t-test.

**Cell migration and invasion assays**. We assayed the migration properties of stable KD cells using OrisTM Universal Cell Migration Assembly Kit (Platypus Technologies). Briefly, stable KD or control Caki-2, 786-O, and A549 cells were seeded into a 96-well plate with Oris Cell Seeding Stoppers at $1.3 \times 10^4$, $1 \times 10^4$, and $1.7 \times 10^4$ cells per well, respectively. To create the detection area, the stoppers were removed after 16 h; stoppers were left in place for the reference wells ($t = 0$ pre-migration control) until the results are read. We used EVOS™ FL Auto Imaging System (Life Technologies) to detect cell migration at 8 and 24 h post seeding in case of Caki-2 and 786-O cell lines, and 24 and 48 h in case of A549 cells. The area of pre-migration ($t = 0$) and post-migration ($t = 8$ and 24 h) were calculated for each condition.

Transwell invasion assay was performed using the 24-well plates BD BioCoat Matrigel Invasion Chamber (BD Biosciences) with 8 μm inserts. For transiently transfected cells, we first performed siRNA and scramble sequence transfection, as described previously. Eight hours post-transfection $3.5 \times 10^4$ Caki-2 transfected cells were re-suspended in 1% FBS RPMI-1640 media and seeded into matrigel-coated inserts. For Caki-2 and A549 stable KD cell lines, $3 \times 10^4$ cells were re-suspended in the appropriate medium supplemented with 1% FBS and seeded into the inserts. Lower chambers were filled with 500 μl of complete medium with 10% FBS as a chemoattractant. Invasion chambers were incubated at humidified 5% $CO_2$ incubator at 37 °C for 22 h. Non-migrated cells were scraped from the interior of the inserts by using a cotton-tipped swab. Cells on the lower surface of the membrane were fixed and stained using the Snabb-Diff kit (Labex), according to the manufacturer's instructions. After staining, the inserts were washed twice in distilled water, and then the membrane were removed from the inserts and kept in slides. Invading cells were photographed at ×20 magnification, and the total number of migrated cells was counted using the EVOS™ FL Auto Imaging System (Life Technologies).

**Chromatin oligo-affinity precipitation**. The ChOP assay was performed as described before by (Akhade et al.)[63] with some modifications. For identification of SCAT7 interacting proteins, HeLa cells ($20 \times 10^6$) were UV crosslinked. The crosslinked pellet was obtained by centrifugation at $1000 \times g$ at 4 °C for 10 min. Cells were resuspended in 2 ml of buffer A (3 mM $MgCl_2$, 10miM Tris-HCl, pH 7.4, 10 mM NaCl, 0.5%v/v NP-40, 0.5 mM PMSF and 100 units/ml RNasin) and incubated on ice for 20 min. Nuclei were harvested by centrifugation and resuspended in 1.2 ml of buffer B (50 mM Tris-HCl, pH 7.4, 10 mM EDTA, 0.5% Triton X-100, 0.1%SDS, 0.5 mM PMSF, and 100 units/ml RNasin) and incubated on ice for 40 min. An equal volume of buffer C (15 mM Tris-HCl, pH 7.4, 150 mM NaCl, 1 mM EDTA, 1% Triton X-100, 0.5 mM PMSF, and 100 units/ml RNasin) was then added and incubated on ice for 15 min. Samples were briefly sonicated using a Bioruptor sonicator (Diagenode) for 20 cycles (30 s on, 30 s off at High Pulse). Eight different oligos complimentary to SCAT7 were pooled with a final concentration of 10 μM and then used for the RNA pull down. As a control, a pool of eight oligos in reverse orientation to the SCAT7 targeting oligos was used. The oligos were added to the chromatin solution along with yeast tRNA (100 μg/ml), salmon sperm DNA (100 μg/ml), and incubated overnight at 4 °C. Samples were then incubated with streptavidin agarose beads for 3 h followed by one wash of each Low salt buffer (20 mM Tris-HCl, pH7.9, 150 mM NaCl, 2 mM EDTA, 0.1% SDS, 1% TritonX-100, 0.5 mM PMSF, and 50 units/ml RNasin), and high salt buffer (20 mM Tris-HCl, pH7.9, 500 mM NaCl, 2 mM EDTA, 0.1% SDS, 1% Triton

**Fig. 8** SCAT7 is a potential therapeutic target for different tumor types. **a** Tumor growth in Balb/c nude mice 8 weeks after subcutaneous inoculation of $1 \times 10^6$ Csh or SCAT7-sh2 786-O cells ($n = 8$ each group). **b** Tumors from Balb/c nude mice after subcutaneous injection of $1 \times 10^6$ Csh or SCAT7-sh1 or SCAT7-sh2 A549 cells for 8 weeks ($n = 6$ for each group). In **a** and **b** tumor volumes ($cm^3$) are expressed as mean ± SD, compared with Csh. **c** Tumor growth inhibition (TGI) for A549 subcutaneous Balb/c nude xenografts treated with 60 pmol of SCAT7 antisense oligonucleotides, LNA-1 and LNA-2, respectively, for a total of four injections ($n = 5$ for each group). **d** Average tumor growth of the xenografts treated with control or SCAT7 LNAs was calculated at each injection as follows: Vtx-Vt0. **e** Real-time qPCR of SCAT7 in A549 tumors collected after treatment with Ctrl-LNA, SCAT7-LNA1, and SCAT7-LNA2. The red bar represents a tumor treated with LNA2, but no downregulation was observed. Values are normalized to endogenous GAPDH. **f** Scatter plot showing the correlation between SCAT7 expression in vivo and the tumor volumes. The red dot represents the tumor that had no significant downregulation of SCAT7 upon LNA treatment. **g** Immunohistochemistry images of Ki67 staining and TUNEL assay for A549 xenografts treated with Ctrl-LNA, SCAT7-LNA-1, and SCAT7-LNA-2 (blue: DAPI, green: TUNEL-GFP). **h** Patient-derived xenograft (PDX) of NSG mice models treated with 100 pmol of Ctrl-LNA or SCAT7-LNA1 for a total of five injections ($n = 6$ for each group). **i** Average growth of tumor volumes of the PDX models treated with control or SCAT7 LNAs. **j**, **k** Model depicting the functional involvement of SCAT7 in regulating cancer hallmarks (**j**) and mechanism of FGF/FGFR signaling regulation by SCAT7 (**k**). The statistical significance shown in **b**–**e** and **i** was derived using a two-tailed unpaired Student's t-test. Data are plotted as mean ± SD (*$p \leq 0.05$; **$p = 0.01$– 0.001; ***$p < 0.001$)

X-100, 0.5 mM PMSF, and 50 units/ml RNasin) and three washes of 1 × PBS. The protein complex was eluted from the beads by incubation with PBS + 0.1%SDS with intermittent mixing at 80 °C for 10 min. All biotinylated oligonucleotides used in the ChOP pull-down are listed in Supplementary Data 4. Few changes were done to the above protocol while assessing the occupancy of *SCAT7* at *FGFR2* promoter. These include (1) formaldehyde crosslinking of cells (instead of UV crosslinking), (2) sonication for 40 cycles (instead of 20 cycles) (3) inclusion of BSA (400 µg/ml) in the oligo binding buffer (4) DNA isolation using phenol–chloroform method after elution from the beads. The % input calculations were made considering the percentage of input chromatin used and the Ct values obtained for the target promoter region from input DNA and ChOP DNA as follows:

$$\% \text{ Input} = \% \text{ of starting input fraction} \times 2^{[Ct(input)-Ct(ChOP)]}$$

**Chromatin and RNA immunoprecipitation**. ChIP was performed according to the protocol described in ref. [64] with some modifications. HeLa cells ($20 \times 10^6$) were harvested and crosslinked using 1% formaldehdye for 10 min at room temperature followed by quenching using 0.125 M of glycine for 5 min. The crosslinked pellet was obtained by centrifugation at $1000 \times g$ at 4 °C for 10 min. The cell pellet was resuspended in 1 ml of SDS lysis buffer (0.1% SDS, 0.5% Triton X-100, 20 mM Tris-HCl, pH 8, and 150 mM NaCl, 1 mM PMSF, and 100 units/ml RNasin) and incubated on ice for 30 min and sonicated using a Bioruptor for a total of 40 cycles (30 s on, 30 s off at High Pulse). Insoluble material was removed by centrifugation at $13000 \times g$ at 4 °C for 10 min. Sonicated DNA was enriched in the range of 100–500 bp. The lysate was incubated with the respective antibodies for immune-precipitation overnight at 4 °C. An aliquot of 4 µg of each antibody was used per 1 mg of lysate for immune-precipitation. The immune-complexes were allowed to bind to Protein G/A Dynabeads for 3 h at 4 °C. The immune-complexes bound to beads were obtained by magnetic precipitation followed by one wash of each low salt buffer (0.1% SDS, 1% Triton-X 100, 2 mM EDTA, 20 mM Tris-HCl, 150 mM NaCl, 0.5 mM PMSF, and 50 units/ml RNasin) and high salt buffer (0.1% SDS, 1% Triton-X 100, 2 mM EDTA, 20 mM Tris-HCl, 500 mM NaCl, 0.5 mM PMSF, and 50 units/ml RNasin). The immune-precipitated material was eluted from the beads by adding 400 µl of elution buffer (100 mM NaHCO₃, 1% SDS, 0.5 mM PMSF and 50 units/ml RNasin) and the samples were incubated at 55 °C for 30 min. The eluates were then processed for DNA isolation by phenol–chloroform method. RIP was performed as described earlier[23]. The eluates from the RIP were processed for total RNA isolation by Trizol method. A aliquot of 250 ng of RNA was used for cDNA synthesis. The % input calculations were made considering the percentage of input chromatin used and the Ct values obtained for the target promoter region from input DNA and ChIP DNA as follows:

$$\% \text{ Input} = \% \text{ of starting input fraction} \times 2^{[Ct(input) - Ct(ChIP)]}$$

**Droplet digital PCR**. Droplet digital PCR was performed using the One-Step RT-ddPCR Advanced Kit (BioRAD #1864021) as per the manufacturer's instructions. The PCR was performed in CFX96 thermal cycler and quantification was done with QX200 droplet reader.

**Mouse xenografts and PDX models**. KIRC and LUAD xenograft models were generated using stable KD cells. 786-O and A549 stable KD cells or control cells were re-suspended in cold PBS with 20% matrigel. We engrafted $1 \times 10^6$ cells subcutaneously in the right flank of 6-week-old female Balb/C nude mice (Janvier labs). Eight weeks post-engrafting, we dissected and measured tumors volumes using the following formula: (long side) × (short side)²/2. For the treatment of LUAD tumors with LNA, we engrafted $1 \times 10^6$ wild-type A549 cells sub-cutaneously. We began treatment with in vivo grade LNA GapmeR molecules at a final concentration of 0.1 mg/Kg when the outside volume of the majority of the tumors reached around 0.5 cm³. We used four mice per group. After terminating the experiment, we dissected the tumors and collected blood samples, livers, spleens, and kidneys from all mice. We obtained the immunocompromised NSG mice models engrafted with patient-derived tumors from the PDX Live™ library at The Jackson Laboratory (Model ID: TM00302). We validated the expression of *SCAT7* in the obtained xenografts by analyzing the RNA-seq data of the original early passaged tumors provided confidentially by The Jackson Library. The mice were kept for one week to acclimatize prior to the LNA injection. For each group ($n = 6$), we injected either 100 pmol of scrambled LNA or *SCAT7* LNA1 every three days for a total period of 15 days. The tumor volumes and growth rates were calculated using the same previously mentioned formula. The animal study protocol was reviewed and approved by the Animal Ethical Review Board, University of Gothenburg, Sweden (Ethical permit no. 45–2015).

**Antibodies and raw Western blots**. Uncropped scans of all western blots are shown in Supplementary Figs 9 and 10. The list of all antibodies used along with their catalog number and dilutions used is provided as Supplementary Data 5.

**Data availability**. The data associated with this publication have been deposited in GEO: GSE92250. All other data are available from the authors upon reasonable request.

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

## Acknowledgements

The results shown here are in whole or part based upon data generated by the TCGA Research Network "http://cancergenome.nih.gov/". The computations for TCGA dataset were performed on resources provided by Uppsala Multidisciplinary Center for Advanced Computational Science (UPPMAX) high-performance computing (HPC) which is part of Swedish National Infrastructure for Computing (SNIC). We thank Prof. Erik Larsson Lekholm for his help with TCGA data. We thank Dr. Leo Kurian and Deniz Bartsch (University of Cologne, Germany) for help with mass spectrometry. This work was supported by the grants from the Knut and Alice Wallenberg Foundation (KAW) (Dnr KAW 2014.0057), Swedish Foundation for Strategic Research (RB13-0204), Swedish Cancer Research foundation (Cancerfonden: Kontrakt no. 150796), the Swedish Research Council (Dnr no: 2017-02834), Barncancerfonden (PR2016-0057), Ingabritt Och Arne Lundbergs forskningsstiftelse, RNATRAIN-Marie-Curie ITN (EU FP7), and LUA/ALF to C.K.

## Author contributions

C.K., M.M.A., V.S.A., S.T.K. and S.S. have conceptualized the work, designed experiments, and wrote the paper. M.M.-F. performed Nascent RNA preparation. S.S has performed nascent RNA-seq analysis, TCGA data analysis and RNA-seq analysis for all siRNA knockdown experiments. S.T.K. has performed all clinical investigations from TCGA data sets. M.M.A. has performed experiments on the SCATs, analyzed RNA-seq on selected SCATs, and helped in experiments with xenografts, derived from cell lines, and patient-derived tumors (PDX). V.S.A. has performed all Western blots, ChOP, RIP, and ChIP experiments. L.S. has performed all experiments on xenografts derived from cell lines and patient-derived tumors (PDX) and wrote the paper. T.M. has provided help in conceptual and experimental design. J.A. helped in clinical investigations.

## Additional information

**Competing interests:** C.K., M.M.A., V.S.A., S.T.K., S.S. and L.S. have filed a provisional patent (no:1750724-5/160080SE) titled "Long noncoding RNA in Cancer" based on findings from the current manuscript. The remaining authors declare no competing interests.

