## [Peer Review File · Nature Communications]

Reviewers' comments:

Reviewer #1 (Remarks to the Author):

Recent studies have highlighted the importance of long ncRNAs (lncRNAs) in several crucial molecular processes, including transcription, genome imprinting and dosage compensation. lncRNAs are known to display differential expression during cell proliferation, cell cycle and cell differentiation and to participate in these processes. Finally, deregulation of lncRNAs is observed under several disease conditions such as cancer. In the present study authors have tried to address the potential involvement of S phase-induced lncRNAs in cancer progression. By performing nascent RNA-seq from HeLa cells that are synchronized in S phase, they identified ~1100 S-phase-induced lncRNAs. Further, by conducting bioinformatics approaches, authors observed that ~60% of S phase-induced lncRNAs showed differential expression in at least two types of the cancer samples. They observed that several hundreds of these lncRNAs have the potential to act as biomarkers with high prediction accuracy. Down regulation of 8 of the candidate lncRNAs in specific cell lines resulted in decrease in the cancer properties, indicating their potential involvement in cancer progression. To gain mechanistic insights into the mode of action of S phase-induced lncRNAs, authors have conducted molecular characterization of SCAT7. Molecular studies indicate that SCAT7 facilitates the recruitment of HnRNP K/YBX RNP complex to the promoters of several of the FGFR genes thereby regulating their transcription. Based on this, authors argue that SCAT7 regulates FGF signaling and participates in the FGF-mediated tumor initiation and progression.

Authors have performed a large number of experiments to cover aspects of lncRNA expression as well as their involvement in tumor progression. Especially, the part of the study, which describes the identification of S phase-induced lncRNAs and the following bioinformatics analyses to determine their involvement in cancer, looks interesting (Figs 1-5). However, experiments performed to determine the role of SCAT7 in FGF signaling though looks promising is incomplete, and require more thorough and careful experimentation (please see below for more details). More importantly, authors have not demonstrated the significance or relevance of S phase specific induction of candidate lncRNAs in their described molecular function and disease connection. Recent studies have shown that depletion of a large number of lncRNAs results in proliferation defects (see the papers by PMIDs: 27798563 and 27980086). But this does not mean that all of these genes directly control these processes. In case of the present study, authors will have to demonstrate that the described molecular function of a particular candidate lncRNA such as SCAT7 happens specifically during S phase of the cell cycle.

Specific comments:

1. Authors have demonstrated that ~1100 lncRNAs are induced during S phase. It is possible that a specific lncRNA could be induced during S phase to produce very unstable lncRNA. Under such circumstances, the regulated transcription from the gene locus rather than the lncRNA transcript itself might play the regulatory role. In that sense, it is not obvious to me the rationale of identifying lncRNAs that are induced during S phase. If one is interested to characterize lncRNAs that function during S phase, it is crucial to identify the candidates that show elevated levels (in terms of steady state levels) during S phase. In this sense, authors should in parallel show the steady state levels of these RNAs (SCAT1-8) during early mid and late S phase stages, by using total RNA as the starting material. In addition, they should also determine the stability of these transcripts during S phase.
2. In addition to double thymidine assay, authors should synchronize cells to S-phase using alternate measures, preferably using assays, which do not use drugs (such as elutriation or serum reactivation) and confirm that the candidate RNAs show S-phase specific expression.
3. Did authors tested whether common lncRNAs presented in figure 2B show similar methylation pattern in those patient samples? This would help one to pinpoint common mode of gene regulation

employed by these lncRNAs in order to regulate their expression in tissues of common origin.

4. Please provide data showing S-phase induced expression of SCAT4, 5, 7 & 8 in KIRC-derived cells such as Caki-2 and for SCAT1 in A549 cells.

5. On what basis, authors conclude that SCAT7 depletion show accumulation of cells in G1/S phase rather than G1 phase. PI-flow cytometry analysis is not a good measure to discriminate between these two phases. Authors should perform alternate molecular experiments to support their claim.

6. Based on the data, I only see a small increase in G1 population in SCAT7-depleted Caki2, A549 and 876-O cells. I am not certain whether such a small change could be used to argue for an S-phase specific function for SCAT7 lncRNA.

7. I am surprised to see senescent like phenotype in SCAT7-depleted HeLa cells. HeLa cells being positive for HPV E6/E7 antigens are extremely refractory to cellular senescence. Earlier studies indicate that repression of E6/E7 antigens (by overexpressing E2 protein) is required for activating senescence. Does SCAT7 depleted cells show E6/E7 repression?

8. Authors need to provide data showing the basic characterization of SCAT7 in the tested cell lines, including northern blot to show the size of the transcript, RNA copy number analyses, data showing noncoding potential of the transcript, SCAT7 RNA localization, polyA status and sequence conservation among vertebrates.

9. Down regulation genes involved in cell proliferation, cell adhesion, cellular senescence, cell migration and in SCAT7-depleted cells could be an 'effect' rather than cause of SCAT7-depletion. In one possible way to test this would be to show that the deregulation of the candidate genes occur prior to the manifestation of phenotype in SCAT7-depleted cells.

10. In similar lines, none of the data presented in figure 6 supports direct role of SCAT7 in regulating FGF signaling. The observed phenotypes (such as cell proliferation and tumor progression defects) observed in SCAT7-depleted cells are so broad that it is nearly impossible to pin point a specific pathway (like FGF) to be responsible for the SCAT7-mediated function. Based on the gene expression data, any dominant pro-tumorigenic signaling pathway could rescue SCAT7 -mediated phenotypes. That does not mean that SCAT7 directly regulates that particular pathway. In order to understand the mechanism exerted by SCAT7, it is vital to determine the copy number of SCAT7. For example, a low copy transcript could regulate the expression of one or few genes, which are part of a bigger signaling network. Identification those genes, which could be the immediate or direct targets of SCAT7 would be critical to determine the function of SCAT7.

11. Is SCAT7 and FGFR2 and FGFR3 genes locate in the same chromosome? Or does it influence FGFR2 & 3 transcription in trans?

12. In order to test the transcriptional status of FGFRs in control and SCAT7-depleted cells, authors should look at RNA pol II occupancy and transcriptional activity in these cells.

13. Does SCAT7 facilitate the proliferation of only tumor cells or does it also regulates cell proliferation of non-tumor cells? I am wondering why SCAT7 ASO-treated mice did not show any defects in the proliferation of non-tumor cells. Does ASOs target only the human specific SCAT7?

14. Does SCAT7 influence the expression of RNPEP, ELF3 and other genes that are located genomic neighborhood?

15. Based on the results that are presented in the ms, I see two important conclusions. One is that these lncRNAs are induced specifically during S phase. Secondly, several of these lncRNAs are deregulated in cancer cells. In the case of SCAT7 authors describe its potential role in regulating FGFR transcription in renal, lung and liver cancer cells. However, it is not clear whether the S phase specific induction of SCAT7 plays any role in regulating FGFR transcription. Does FGFR genes in these cells also induce only during S phase? Does depletion of SCAT7 in cell types that do not show S phase specific induction of SCAT7 also display defects in FGFR transcription and downstream cancer phenotypes? These questions need to be addressed to make the story coherent.

16. Authors have shown S phase specific induction of SCAT7 in HeLa cells. In addition data described in figs 6 & 7 demonstrate functional role of SCAT7 in regulating FGFR transcription in HeLa cells. Does the cervical cancer samples show deregulation of SCAT7? In similar lines, does SCAT7 and FGFR genes

show positive correlation in patient samples, which show elevated levels of SCAT7?

17. Please make the corrections in lines 317-319, page 11. Stable KO of SCAT7.....increased caspase 3/7 activity, migration and invasive capacity". Based on the data (5E & F), SCAT7-depleted cells show reduced migration and invasion.

18. Please add error bars to data presented in S5M.

19. Page number 4, lane 110: please also add the following references PMIDs: 27798563 and 27980086.

Reviewer #2 (Remarks to the Author):

In this tour-de-force investigation, the authors set out to identify S-phase-specific long noncoding RNAs (lncRNAs) expressed in cancer using a method to isolate S-phase RNAs selectively. After high-throughput identification of lncRNAs associated with proliferative phases of the cell division cycle, and analysis of TCGA dataset, they focused on 8 lncRNAs that showed preferentially elevated expression in the Pan-Cancer TCGA dataset and predicted patients' survival in multiple cancers. They termed these lncRNAs S-phase Cancer-Associated Transcripts (SCATs) and set out to perform collective functional validation of SCAT lncRNAs and robust mechanistic molecular analysis of one of the SCAT lncRNAs, SCAT7. SCAT7 was further shown to be involved in cell proliferation, invasion, protection against apoptosis, anchorage-independent growth, and evasion of cell senescence. Although the manuscript is excellent, although the authors are encouraged to consider a few suggestions.

Comments

1. Give that SCAT genes are Are there E2F promoter elements in the SCAT genes?
2. If HRNPK and YBX1 are silenced, are the effects of SCAT7 on proliferation, apoptosis, and senescence reversed? Could the authors monitor a handful of endogenous HRNPK/YBX1 target genes after modulating SCAT7 levels, to solidify the idea that SCAT7 influences this transcriptional program?
3. The text reads well in general, but needs some editing to correct typos (e.g., 'Principal Component Analysis', not 'Principle Component Analysis') as well as for grammar and syntax.

Reviewer #3 (Remarks to the Author):

The authors profiled the S-phase specific lncRNAs and identified lots of candidates that are differentially expressed across cancer types and can potentially serve as prognostic biomarkers. Although the regulatory roles of lncRNAs in S-phase have not been fully investigated, this study provides a rich resource for our understanding of S-phase specific lncRNAs. Furthermore, the authors investigated the potential mechanism of a lncRNA, SCAT7, which may promote tumor progression by alternating signaling pathways relating to the hallmarks of cancer. The role of SCAT7 in tumorigenesis was validated at molecular level, in multiple cell lines and through mouse xenografts. Several issues need to be addressed as follows.

Major points

1. In the first part of the Results section, the authors showed that the EtU labeling is efficient for isolation S-phase transcripts. However, in Figure 2B, the expression pattern of S phase-T2 specific lncRNAs is much more close to the expression pattern of unsynchronized cells compared to S-phase T1 and T3. Then in Figure 2C, the authors showed that 4 cluster of lncRNAs with distinctive temporal

expression patterns across the S-phase T1, T2, T3 and unsynchronized cells. Have the authors investigated the functional characteristics of these 4 cluster or any biological processes that is related to the temporal expression pattern?

2. In the first step of experiment design, the authors selected 1,145 lncRNAs showing significant temporal expression patterns in HeLa cells across S-phase compared to unsynchronized cells as 'S-phase specific' lncRNAs. However, the TCGA samples are unsynchronized cells and may mostly consist of G1 phase cells according to the cell population in cancer cell lines in this study. Hence, lncRNAs that are significantly highly expressed in TCGA samples are not necessarily S-phase specific and may depend on the cancer type. These S-phase specific lncRNAs are a subset of expressed lncRNAs in HeLa cells and should not be considered as S-phase specific lncRNAs across cancer types. It is better to show the difference between these S-phase specific lncRNAs and other differentially expressed lncRNAs across cancer types in terms of expression patterns and targets, etc.

3. The Methods part of the paper is poorly written in the main text. Some of them are scattered in the supplementary materials. The authors need to describe them clearly in the paper, especially various methods used for data analyses.

4. The data need to more carefully analyzed. For instance, to analyze the function of the S-phase specific lncRNAs, they used the neighboring protein-coding RNAs within 50kb window. The reason of 50Kb need to be explained. Different sizes of windows may be used to check if the enrichments of certain functional terms are robust.

5. The authors used "data not shown" in multiple places, which should be avoided. Data, Tables and figures (maybe as supplementary materials) need to be provided.

Minor points

1. The authors mentioned bioinformatics tools such as Short Time-series Expression Miner (STEM) clustering and GeneSCF. Maybe a brief introduction of the purpose and advantages of the tools and description of the results should be included to help the reader to understand the results.

2. Table S1 contains multiple tables of different categories and it is not clear which of Table S1 is referred to in the main text. Maybe a summary and a brief description should be included along with the tables or refer the tables in the Excel file as different numberings (e.g. Table S1, Table S2, Table S3 ...).

3. Why two bands in Figure7G of FGFR2 and ERK1/2? Are there some modifications?

4. In line 311, they declare that "H2228 and HepG2 cell lines were too sensitive to long term SCAT7 downregulation using shRNA particles." Some supporting references and/or data are needed.

We thank all the reviewers for providing comments/suggestions on our manuscript

Reviewer #1 (Remarks to the Author):

Recent studies have highlighted the importance of long ncRNAs (lncRNAs) in several crucial molecular processes, including transcription, genome imprinting and dosage compensation. LncRNAs are known to display differential expression during cell proliferation, cell cycle and cell differentiation and to participate in these processes. Finally, deregulation of lncRNAs is observed under several disease conditions such as cancer. In the present study authors have tried to address the potential involvement of S phase-induced lncRNAs in cancer progression. By performing nascent RNA-seq from HeLa cells that are synchronized in S phase, they identified ~1100 S-phase-induced lncRNAs. Further, by conducting bioinformatics approaches, authors observed that ~60% of S phase-induced lncRNAs showed differential expression in at least two types of the cancer samples. They observed that several hundreds of these lncRNAs have the potential to act as biomarkers with high prediction accuracy. Down regulation of 8 of the candidate lncRNAs in specific cell lines resulted in decrease in the cancer properties, indicating their potential involvement in cancer progression. To gain mechanistic insights into the mode of action of S phase-induced lncRNAs, authors have conducted molecular characterization of SCAT7. Molecular studies indicate that SCAT7 facilitates the recruitment of HnRNP K/YBX RNP complex to the promoters of several of the FGFR genes thereby regulating their transcription. Based on this, authors argue that SCAT7 regulates FGF signaling and participates in the FGF-mediated tumor initiation and progression.

Authors have performed a large number of experiments to cover aspects of lncRNA expression as well as their involvement in tumor progression. Especially, the part of the study, which describes the identification of S phase-induced lncRNAs and the following bioinformatics analyses to determine their involvement in cancer, looks interesting (Figs 1-5). However, experiments performed to determine the role of SCAT7 in FGF signaling though looks promising is incomplete, and require more thorough and careful experimentation (please see below for more details). More importantly, authors have not demonstrated the significance or relevance of S phase specific induction of candidate lncRNAs in their described molecular function and disease connection. Recent studies have shown that depletion of a large number of lncRNAs results in proliferation defects (see the papers by PMIDs: 27798563 and 27980086). But this does not mean that all of these genes directly control these processes. In case of the present study, authors will have to demonstrate that the described molecular function of a particular candidate lncRNA such as SCAT7 happens specifically during S phase of the cell cycle.

Specific comments:

1. Authors have demonstrated that ~1100 lncRNAs are induced during S phase. It is possible that a specific lncRNA could be induced during S phase to produce very unstable lncRNA. Under such circumstances, the regulated transcription from the gene locus rather than the lncRNA transcript itself might play the regulatory role. In that sense, it is not obvious to me the rationale of identifying lncRNAs that are induced during S phase. If one is interested to characterize lncRNAs that function during S phase, it is crucial to identify the candidates that show elevated levels (in terms of steady state levels) during S phase. In this sense, authors should in parallel show the steady state levels of these RNAs (SCAT1-8) during early mid and late S phase stages, by using total RNA as the starting material. In addition, they should also determine the stability of these transcripts during S phase.

In our study, we have employed a nascent RNA capture assay coupled with deep sequencing to identify the transcriptional events that occur primarily during the S phase progression. There is a growing body of evidence supporting the higher resolution and sensitivity of the nascent RNA sequencing approach over the conventional steady-state RNA sequencing in detecting the transcriptome temporal dynamics (**Core et al., 2008; Danko et al., 2015; Hah et al., 2011; Hah et al., 2013; Kwak et al., 2013; Nojima et al., 2015; Sun et al., 2015**). Moreover, we have successfully utilized the nascent RNA capture assay to provide insights on the temporal separation of DNA replication and RNA transcriptional events during the S-phase which was not evident using the total RNA sequencing strategy (**Meryet-Figuere et al., 2014**). In the current manuscript, we have already utilized both total RNA sequencing (referred to as unlabeled samples) and nascent RNA sequencing (referred to as labeled samples) to determine the exact temporal patterns of RNA transcription during different compartments of the S-phase

(Supplementary Table 1). As shown in Fig. 1b, our strategy provided a better separation of the ongoing transcriptional events in the S-phase compared to the total RNA sequencing.

To provide functional evidence on the role of the identified lncRNAs, we selected eight lncRNAs which were enriched in the synchronized samples (T₁, T₂ and T₃) with a log₂ fold change ≥ 1 over the unsynchronized sample. The selected lncRNAs were also enriched in total RNA sequencing. The expression datasets are already provided in Supplementary Table 1 and the analysis is described in the Supplementary Materials and Methods. The table below shows the log₂ fold change of the selected SCATs.

lncRNA	Log ₂ FC in labeled samples			Log ₂ FC in unlabeled samples		
	T ₁	T ₂	T ₃	T ₁	T ₂	T ₃
SCAT1	0,631441205	1,753858529	1,826127821	0,801893781	1,122543482	0,797403526
SCAT2	1,368406799	1,016892934	-0,75883468	-0,520034315	1,122543482	-0,202596474
SCAT3	1,915894594	0,35598158	1,222549642	0,31490644	0,57605513	-0,202596473
SCAT4	0,631441204	1,239285355	1,24116532	1,21693128	0,759973404	-0,202596474
SCAT5	3,913476572	2,464351911	1,911016719	1,527271401	1,840143752	0,844709241
SCAT6	2,011713286	1,32501523	1,643263764	3,278707478	2,952618481	2,450845765
SCAT7	1,472743459	2,046640278	2,148055916	3,546054877	3,747034347	3,189720949
SCAT8	4,472743459	4,476324553	4,048520242	2,649890687	3,625043823	2,119331621

As per the reviewer’s suggestion, for further confirmation, we performed a quantitative real-time PCR validation of the expression levels of the selected SCATs using the total RNA collected from synchronized and unsynchronized samples. The total RNA validation demonstrated an enrichment of the eight SCATs during S-phase over the unsynchronized sample. The stability of the selected lncRNAs can be inferred from both RNA sequencing and qPCR validation showing a dynamic change during different time points. However, to avoid any confusion, in the revised manuscript we termed the identified lncRNAs as “S-phase enriched” instead of “S-phase specific”. The new validation data are provided as a Supplementary Fig. 1d.

Figure legend: RT-qPCR validation of the expression levels of the selected SCATs at different time points post-synchronization in HeLa cells using total RNA.

2. In addition to double thymidine assay, authors should synchronize cells to S-phase using alternate measures, preferably using assays, which do not use drugs (such as elutriation or serum reactivation) and confirm that the candidate RNAs show S-phase specific expression.

We appreciate the reviewer's concern regarding the reproducibility of the identified lncRNAs' temporal expression during S-phase. Although the use of reversible chemical inhibitors of DNA synthesis is the most favorable choice for efficient cell synchronization (Davis et al., 200; Fox et al., 1987; Ma and Poon, 2011; Petermann et al., 2010), we have also confirmed the expression of the eight SCATs in HeLa cells using the serum starvation method. We observed an enrichment of the SCATs (except SCAT3 which matches our RNA-seq data where SCAT3 shows S-phase enrichment only in EtU labeled sample but not unlabeled sample) over the unsynchronized cells. Please, refer to the response to the first comment.

3. Did authors tested whether common lncRNAs presented in figure 2B show similar methylation pattern in those patient samples? This would help one to pinpoint common mode of gene regulation employed by these lncRNAs in order to regulate their expression in tissues of common origin.

We have investigated the methylation status of all cancer associated S-phase expressed lncRNAs presented in Fig. 2a which also includes lncRNAs from Fig. 2b. We did not find any common lncRNA from Fig. 2b with significant methylation status. We could only find 55 lncRNAs (Fig. 2c, Supplementary Fig. 2a and Supplementary Table 1) showing methylation status with the criteria mentioned in the methylation analysis ("differential methylation was considered significant only if a particular pattern, hypo- or hypermethylation was supported by at least 10 patient samples and with null patients opposing the pattern").

4. Please provide data showing S-phase induced expression of SCAT4, 5, 7 & 8 in KIRC-derived cells such as Caki-2 and for SCAT1 in A549 cells.

The focus of the current study emphasizes the oncogenic roles of the lncRNAs identified through a cell cycle-based screening in HeLa cells. We utilized our cell cycle-based RNA-seq datasets and applied multiple filtering criteria as outlined in a workflow (Fig. 1e) to select potential candidates for further functional investigations in different cancer cell line model systems. However, as the reviewer suggested, we checked the expression levels of SCAT4, SCAT5, SCAT7 and SCAT8 in Caki-2 cells. Although the synchronization of Caki-2 cells has been reported earlier to be inefficient (Bongiorno-Borbone et al., 2008), we synchronized Caki-2 cells using serum starvation. As previously shown (Poplawski and Nauman, 2008), the serum starved Caki-2 cells progress through G1, G1/S and S phases at 18h, 22h and 26h post-synchronization, respectively. We selected three time points post synchronisation: 4h (T1), 8h (T2) and 24h (T3). The qPCR validation of the selected SCATs demonstrated an increase in their expression levels following serum reactivation. We have included the new data as Supplementary Fig. 3j.

Figure legend: Quantitative PCR validation of the expression levels of the selected SCATs at different time points post-synchronization in serum-starved Caki-2 cells using the total RNA. SF-T0 indicates Serum Free, 0 hrs post serum reactivation.

Similarly, we have synchronized A549 cells using serum starvation for 48h as previously described. Since the S-phase in A549 cells lasts 12-16 hours as indicated by our cell cycle profiling and previous reports (**Knox et al., 2000; Lauand et al., 2015; Zhang et al., 2008**), we collected RNA at 4h, 6h and 18h following the serum reactivation. According to the qPCR validation, *SCAT1* showed an elevated expression 6 hours following the serum reactivation. We have included this data as **Supplementary Fig. 4g**.

Figure legend: Quantitative PCR validation of the expression levels of *SCAT1* and *SCAT7* at different time points post-synchronization in serum-starved A549 cells using the total RNA.

5. On what basis, authors conclude that *SCAT7* depletion show accumulation of cells in G1/S phase rather than G1 phase. PI-flow cytometry analysis is not a good measure to discriminate between these two phases. Authors should perform alternate molecular experiments to support their claim.

We thank the reviewer for this important suggestion. In the revised manuscript, based on both propidium iodide and DAPI staining, we have referred to the cell cycle perturbations associated with SCATs knockdown as G1 phase accumulation rather than G1/S. Moreover, in order to reinforce the function of *SCAT7* in the S-phase progression, we have performed a Click-iT[®] chemistry-based incorporation of the 5-ethynyl-2'-deoxyuridine analogue (EdU) followed by a fluorescent detection of the Alexa Fluor[®] moiety. In comparison to the control samples, the *SCAT7* silencing has drastically decreased the DNA synthesis as represented by the diminishing of the green fluorescence signal of the incorporated EdU analogue. The decrease of the DNA synthesis shown by the EdU incorporation assay is consistent with the flow cytometry-based cell cycle profiling of *SCAT7* knockdown in HeLa, Caki2 and A549 cell lines. The EdU incorporation data are included in the revised manuscript as **Supplementary Fig. 5h**.

Along similar lines, the transcriptome analysis of *SCAT7* knockdown cells showed a significant deregulation of some of the well-known cell cycle-associated genes. For instance, *CDKN1A* was significantly upregulated at the transcriptional level upon *SCAT7* silencing (**Supplementary Table 4**). In addition, *SCAT7* silencing decreases *Cyclin D1* (*CCND1*) at both transcriptional and translational levels (**Fig. 6e and Supplementary Table 4**). Additionally, as shown in **Fig. 6e**, *SCAT7* silencing drastically decreased the phosphorylation of the tumor suppressor gene retinoblastoma (Rb) and thus inducing G1 accumulation and cell cycle arrest.

Figure legend: EdU incorporation assay in HeLa, Caki-2 and A549 cells showing a drastic decrease of DNA synthesis indicated by Alexa Fluor 488 signal following *SCAT7* depletion.

6. Based on the data, I only see a small increase in G1 population in *SCAT7*-depleted Caki2, A549 and 876-O cells. I am not certain whether such a small change could be used to argue for an S-phase specific function for *SCAT7* lncRNA.

The silencing of *SCAT7* in HeLa, Caki2 and HepG2 cells induced $\geq 10\%$ accumulation at G1 phase while A549, 786-O and MCF-7 cells showed less accumulation at G1 phase. However, in all tested cell lines, the *SCAT7* knockdown has significantly reduced the S-phase progression by more than 50% as indicated by the flow cytometry analysis. One possible explanation for the lack of accumulation of *SCAT7* depleted cells in G1 corresponding to the percentage of S-phase arrested cells may be due to channeling of most of the S-phase arrested cells into apoptosis. This possibility supported by the observation that significant accumulation of apoptotic cells upon *SCAT7* depletion in all the analysed cell types (please see figure 5D). Further, to support our observations on the role of *SCAT7* in DNA synthesis, we performed the EdU incorporation assay in HeLa, Caki2 and A549 cell lines. As indicated in our response to the point number 5, the obtained results further confirm the vital function of *SCAT7* in DNA synthesis during the S-phase. Moreover, our data come in line with recent reports that describe the role of different lncRNAs in regulating cell cycle-related processes in a same manner. For example, the silencing of CONCR lncRNA (presented in our manuscript as *SCAT4*) has been shown to induce cell cycle perturbations in A549 cells mostly associated with S-phase inhibition and G1 accumulation in a percentage similar to *SCAT7* (Marchase et al., 2016). The oncogenic lncRNA MALAT1 has also been shown to control the cell cycle progression of human diploid lung fibroblasts (WI-38) by regulating the expression of the transcription factor B-MYB (Tripathi et al., 2013). The transcriptional modulation of MALAT1 in WI-38 cells induces G1 accumulation and $> 50\%$ reduction of the S-phase as measured by BrdU-PI flow cytometry in a manner similar to *SCAT7*. Similarly, the silencing of FAL1, HOXA11-AS lncRNAs exhibits effects on cell cycle progression in MCF-7 cells and multiple human glioblastoma cell lines, respectively, similar to that seen with *SCAT7* (Hu et al., 2014; Wang et al., 2016).

7. I am surprised to see senescent like phenotype in *SCAT7*-depleted HeLa cells. HeLa cells being positive for HPV E6/E7 antigens are extremely refractory to cellular senescence. Earlier studies indicate that repression of E6/E7 antigens (by overexpressing E2 protein) is required for activating senescence. Does *SCAT7* depleted cells show E6/E7 repression?

As the reviewer indicated in the above comment, the induction of cellular senescence in HeLa cells has been addressed extensively in previous reports (Goodwin et al., 2000; Goodwin and DiMaio, 2001; Hall and Alexander, 2003; Kang et al., 2004). In our study, the RNA-seq analysis and the subsequent pathways enrichment analysis indicated that *SCAT7* depletion in HeLa cells affected several genes involved in regulating the Senescence-Associated Secretory Phenotype (SASP) pathway (Supplementary Table 4). Interestingly, *SCAT7* depletion has also affected the DNA Damage/Telomere Stress Induced Senescence pathway which is documented as a major pathway involved in senescence. The subsequent qPCR validation of different senescence-associated markers such as *CDKN1A*, *CDKN2A*, *IL-6* and *IL-8* has further strengthened our data. Finally, as mentioned in earlier reports, E6/E7 repression leads to an alteration in the phosphorylation pattern of retinoblastoma protein (Rb). In the presented Figure 6e, *SCAT7* silencing induced a significant dephosphorylation of Rb which led to the activation of Rb as a negative regulator of E2F factors to impose a cell cycle arrest and cellular senescence. Thus, we conclude that

SCAT7 expression is required to repress the activity of Rb and the associated senescence to sustain the cell proliferation.

8. Authors need to provide data showing the basic characterization of *SCAT7* in the tested cell lines, including northern blot to show the size of the transcript, RNA copy number analyses, data showing noncoding potential of the transcript, *SCAT7* RNA localization, polyA status and sequence conservation among vertebrates.

As per the reviewer’s suggestion, we confirmed the transcript size using Northern blotting which demonstrated the estimated size of *SCAT7* isoforms. We included Northern blot in the supplementary information as **Supplementary Fig. 5a**. Moreover, since *SCAT7* is already an annotated transcript, we amplified, cloned the full-length sequence and confirmed the size by Sanger sequencing. We calculated the RNA copy number of *SCAT7* using digital droplet PCR (ddPCR) in HeLa and A549 cancer cells, revealing 224 and 496 copies/ng RNA (A549 cells known to have >2500 copies *MALAT1* and *NEAT1* respectively, Dodd et al.,2013).

We checked the noncoding potential of *SCAT7* using two different tools. As indicated below, *SCAT7* isoforms don’t show any coding capacity.

SCAT7 variants	CPAT_probability	CPAT_coding_label	CPC_score	CPC_status
RP11-465N4.4-001	0.060749961	no	-0.979115	noncoding
RP11-465N4.4-002	0.103533508	no	-0.842284	noncoding

In CPAT, the probability < 0.364 indicates a noncoding sequence while the negative values in CPC score indicate a noncoding potential.

The localization of *SCAT7* in HeLa cells was already provided in the **Supplementary Fig. 7a**.

SCAT7 is a polyadenylated transcript as we amplified *SCAT7* transcripts in our RT-qPCR reactions using both oligo dT primer and random hexamer primers, independently. Also, we cloned the full length transcript of *SCAT7* utilizing the polyA tail which was further confirmed by Sanger sequencing.

We have also checked the sequence conservation of *SCAT7* among the vertebrates. As shown below *SCAT7* doesn’t show any sequence homology as determined by phastCons score and phyloP, and is expressed only in human cells. However, the neighbouring protein-coding genes are conserved.

In the revised manuscript we have included additional figures (Supplementary **Figs. 5a and b**) and few sentences describing the noncoding capacity, polyadenylation status and sequence conservation of *SCAT7*.

9. Down regulation genes involved in cell proliferation, cell adhesion, cellular senescence, cell migration and in SCAT7-depleted cells could be an 'effect' rather than cause of SCAT7-depletion. In one possible way to test this would be to show that the deregulation of the candidate genes occur prior to the manifestation of phenotype in SCAT7-depleted cells.

We thank the reviewer for raising very interesting question on lncRNA knockdowns and phenotype manifestation in cancer cell line models. However, it would be difficult to predict the window period between gene expression changes and manifestation of phenotype. In that scenario, it requires a lot of effort to measure temporal transcription events leading to manifestation of phenotype. We believe that this issue is beyond the scope of current investigation. More importantly, in our investigation, we have included several lines of evidence to support the vital functions of SCAT7 in multiple model systems including *in vitro* and *in vivo* models using siRNA, LNA and shRNA. As indicated in the RNA-seq analysis with subsequent RT-qPCR validation (**Supplementary Table 4 and Supplementary Fig. 6b**), the silencing of SCAT7 caused a transcriptome-wide changes affecting the expression levels of many important genes involved in different cancer hallmarks (Please, refer to the graphical summary in **Fig. 8j**). Remarkably, SCAT7 knockdown induced a significant upregulation of the *CDKN1A* and *CDKN2A* genes in many cell lines which are known to induce cell cycle arrest. Also, as shown already in **Fig. 6e**, SCAT7 knockdown reduced significantly the expression level of *CCND1* and the phosphorylation of Rb. The downregulation of SCAT7 induced apoptosis and significantly altered the proliferation capacity, cell migration and invasion in the tested model systems. To complement our results, we performed an ectopic expression of SCAT7 which indeed increased the cell proliferation rates and reversed the senescence-associated phenotypes in the investigated cell lines (**Fig. 5h, Figs. 5m-5p, Fig. 6g and Supplementary Fig. 6g**). Moreover, the downregulation of SCAT7 *in vivo* reduced the proliferation rate of the engrafted tumors as indicated by Ki67 and TUNEL immunostaining, respectively (**Fig. 8g and Supplementary Fig. 8b**). Thus, our presented data strongly indicate that the effects on different phenotypes associated with the cancer hallmarks are direct effects of SCAT7 transcriptional modulation, but not the vice versa.

10. In similar lines, none of the data presented in figure 6 supports direct role of SCAT7 in regulating FGF signaling. The observed phenotypes (such as cell proliferation and tumor progression defects) observed in SCAT7-depleted cells are so broad that it is nearly impossible to pin point a specific pathway (like FGF) to be responsible for the SCAT7-mediated function. Based on the gene expression data, any dominant pro-tumorigenic signaling pathway could rescue SCAT7 -mediated phenotypes. That does not mean that SCAT7 directly regulates that particular pathway. In order to understand the mechanism exerted by SCAT7, it is vital to determine the copy number of SCAT7. For example, a low copy transcript could regulate the expression of one or few genes, which are part of a bigger signaling network. Identification those genes, which could be the immediate or direct targets of SCAT7 would be critical to determine the function of SCAT7.

As per Ct values of SCAT7 and GAPDH in our RT-qPCR analysis and also SCAT7 copy number analysis by ddPCR indicates that SCAT7 has a fair amount of expression (please see our response to comment 8). We are of the opinion that even lowly expressed lncRNAs can have broad functions if they are part of the important transcriptional regulatory chromatin remodeling complexes or regulators of key master genes of cell signaling pathways. Moreover, deletion of highly expressed genes such as *MALAT1* has no seeming effects on the mouse development (Eissman et al., 2012).

Regarding the direct role of SCAT7 in FGF signaling, we have already provided functional data that support the role of SCAT7 in regulating different members of FGF/FGFR signaling pathway. The RNA-seq data presented in **Fig. 6** provide the first line of evidence on the role of SCAT7 in modulating the expression levels of many FGF/FGFR members in different cell lines. Following the RNA-seq analysis, as already shown in **Fig. 7a** and **Supplementary Table 4**, we performed a Chromatin Oligo affinity Precipitation (ChOP) followed by mass spectrometry (LC/MS-MS) analysis of SCAT7 interacting protein. We identified and validated hnRNPK and YBX1 as genuine protein counterparts of SCAT7 (**Figs. 7b, c, l and o**). Mechanistically, we have already shown that SCAT7/hnRNPK/YBX1 complex is enriched at the proximal promoter region of *FGFR2* and *FGFR3* in HeLa cells to promote their transcriptional activity (**Figs. 7h-k**). Similarly, the SCAT7/hnRNPK/YBX1 complex is enriched at the *FGFR3* proximal promoter in A549 cells and facilitates the transcription of *FGFR3* (**Figs. 7l-n**). In Caki-2 cells, SCAT7/hnRNPK complex induces the expression of *FGFR2* (**Figs. 7o-q**). Moreover, the independent downregulation of hnRNPK and YBX1 phenocopied the effect of SCAT7 downregulation (**Figs. 7d-g**). So, as we

have written in the discussion part that the functional conservation of *SCAT7*-hnRNP-K-YBX1 RNP complex in different cancer models presents a mechanistic model of *SCAT7* dependent *FGF/FGFR* regulation. The recruitment of the *SCAT7*-hnRNP-K-YBX1 RNP complex at the promoter regions of *FGFR2* and *FGFR3* promotes transcriptional activation of the FGF/FGFR pathway, leading to sustained cell proliferation via PI3K/AKT and Ras/MAPK pathways.

11. Is *SCAT7* and *FGFR2* and *FGFR3* genes locate in the same chromosome? Or does it influence *FGFR2* & 3 transcription in trans?

According to the recent human genome assembly (GRCh38/hg38) and Gencode annotation v24, *SCAT7* is located at chromosome 1 covering the region from 202,000,094 to 202,010,353. The protein-coding gene *FGFR2* is located at chromosome 10 while *FGFR3* gene is located at chromosome 4. So, our RNA-seq data and the subsequent validations demonstrate the role of *SCAT7* in regulating its target genes *in trans*.

12. In order to test the transcriptional status of FGFRs in control and *SCAT7*-depleted cells, authors should look at RNA pol II occupancy and transcriptional activity in these cells.

As per the reviewer suggestion, we have performed RNA Pol II ChIP in HeLa, A549 and Caki2 cells to look for RNA Pol II occupancy downstream to the TSS of different FGFRs. We observe a significant decrease in RNA Pol II occupancy upon silencing of *SCAT7* in all the three cell lines. We have included this data as **Supplementary Fig. 7d**.

13. Does *SCAT7* facilitate the proliferation of only tumor cells or does it also regulates cell proliferation of non-tumor cells? I am wondering why *SCAT7* ASO-treated mice did not show any defects in the proliferation of non-tumor cells. Does ASOs target only the human specific *SCAT7*?

We apologize for not providing a text to explain the findings presented already in the **Supplementary Fig. 5c**. In the presented figure, we compared the expression levels of *SCAT7* in a normal lung cell line, BEAS-2B obtained from ATCC, with the lung adenocarcinoma cell lines. The qPCR expression analysis showed an elevated expression of *SCAT7* in the cancer cell lines compared to the normal cells which comes in line with our TCGA analysis. We have also investigated the expression level of *SCAT7* in the transformed non-cancerous human embryonic kidney cell line HEK293. The depletion of *SCAT7* reduced the proliferation capacity of HEK293 cells by 40-50% and we have presented this data as **Supplementary Fig. 5g**.

According to the mouse genome assembly (GRCm38/mm10) and gencode annotation, and as shown in our response to point number eight, *SCAT7* is not expressed in mouse cells and doesn't have a sequence homology among the vertebrates. Thus, the localized injection of *SCAT7* ASOs affected only the engrafted human cancerous cells and the patient-derived tissues without interfering with the mouse cells.

14. Does *SCAT7* influence the expression of *RNPEP*, *ELF3* and other genes that are located genomic neighborhood?

By employing siRNAs, shRNAs and LNAs that target the unique non-overlapping exon of *SCAT7* without affecting the flanking genes, we have shown that the *SCAT7* depletion did not affect the expression of neighboring genes in HeLa and Caki-2 cell lines. However, we found significantly reduced *ELF3* expression only in A549 cells (**Supplementary Tables 3a-g**).

15. Based on the results that are presented in the ms, I see two important conclusions. One is that these lncRNAs are induced specifically during S phase. Secondly, several of these lncRNAs are deregulated in cancer cells. In the case of *SCAT7* authors describe its potential role in regulating FGFR transcription in renal, lung and liver cancer cells. However, it is not clear whether the S phase specific induction of *SCAT7* plays any role in regulating FGFR transcription. Does FGFR genes in these cells also induce only during S phase? Does depletion of *SCAT7* in cell

types that do not show S phase specific induction of SCAT7 also display defects in FGFR transcription and downstream cancer phenotypes? These questions need to be addressed to make the story coherent.

We thank the reviewer for the valuable suggestion. We checked the expression levels of *FGFR2*, *FGFR3* and *FGFR4* in the serum-starved synchronized cells. In HeLa cells, the expression of *FGFR2* and *FGFR3* is enriched in the S-phase in a similar manner to *SCAT7*. In A549 cells, *SCAT7* does not show S-phase specific induction. However in A549 cells, both *FGFR3* and *FGFR4* are expressed at later time points of the cell cycle matching the expression patterns of *SCAT7*. In A549 cells, irrespective of *SCAT7* not being induced in S phase specific manner, knockdown of *SCAT7* downregulates the expression of *FGFR3* and *FGFR4*. On the other hand, the expression of *FGFR2* in Caki-2 cells showed an elevated levels in both G1 phase and later time points of the cell cycle which doesn't match the expression patterns of *SCAT7* in this context. So, based on the investigated expression patterns, *SCAT7* contributes to the transcriptional regulation of FGFR members regardless to their temporal expression. Thus, we conclude that the depletion of *SCAT7* in cell types that do not show S phase specific induction of *SCAT7* also display defects in FGFR transcription and downstream cancer phenotypes. We have included this data as **Supplementary Fig. 6d**.

Figure legend: The expression patterns of *FGFR2*, *FGFR3* and *FGFR4* at different time points in HeLa, A549 and Caki-2 cells.

16. Authors have shown S phase specific induction of SCAT7 in HeLa cells. In addition data described in figs 6 & 7 demonstrate functional role of SCAT7 in regulating FGFR transcription in HeLa cells. Does the cervical cancer samples show deregulation of SCAT7? In similar lines, does SCAT7 and FGFR genes show positive correlation in patient samples, which show elevated levels of SCAT7?

According to the TCGA analysis presented in the **Supplementary Table 2b**, *SCAT7* is significantly upregulated (\log_2 fold change, 2.8) in the uterine corpus endometrial carcinoma (UCEC) compared to normal tissues.

As per the reviewer's suggestion, we investigated *SCAT7* and *FGFR2/FGFR3* expression correlation in UCEC, KIRC and LUAD cancers. We found that significant expression correlation either in low or high expression groups, which could be due to *SCAT7* in part contributes to the regulation of FGFR members, and also as in the case of other genes, several spatio-temporally controlled factors may influence FGFR transcriptional regulation. We have included the new findings as **Supplementary Table 3h**.

17. Please make the corrections in lines 317-319, page 11. Stable KO of SCAT7.....increased caspase 3/7 activity, migration and invasive capacity". Based on the data (5E & F), SCAT7-depleted cells show reduced migration and invasion.

We apologize for the typing mistake. We have made the correction in the revised version of the manuscript.

18. Please add error bars to data presented in S5M.

We have added the error bars in the figure.

19. Page number 4, lane 110: please also add the following references PMIDs: 27798563 and 27980086.

We have cited the mentioned references as indicated by the reviewer.

Reviewer #2 (Remarks to the Author):

In this tour-de-force investigation, the authors set out to identify S-phase-specific long noncoding RNAs (lncRNAs) expressed in cancer using a method to isolate S-phase RNAs selectively. After high-throughput identification of lncRNAs associated with proliferative phases of the cell division cycle, and analysis of TCGA dataset, they focused on 8 lncRNAs that showed preferentially elevated expression in the Pan-Cancer TCGA dataset and predicted patients' survival in multiple cancers. They termed these lncRNAs S-phase Cancer-Associated Transcripts (SCATs) and set out to perform collective functional validation of SCAT lncRNAs and robust mechanistic molecular analysis of one of the SCAT lncRNAs, SCAT7. SCAT7 was further shown to be involved in cell proliferation, invasion, protection against apoptosis, anchorage-independent growth, and evasion of cell senescence. Although the manuscript is excellent, although the authors are encouraged to consider a few suggestions.

Comments

1. Give that SCAT genes are Are there E2F promoter elements in the SCAT genes?

As per the reviewer's suggestion, we investigated the enrichment of E2F1 transcription factor binding sites at the promoter regions (\pm 2kb from TSS) of the identified S-phase enriched RNAs in both EtU labeled and unlabeled samples. We found that 72 EtU labeled and 67 unlabeled S-phase lncRNAs promoters were associated with E2F1 binding sites. This information is updated in **Supplementary Table 1d**.

2. If HRNPK and YBX1 are silenced, are the effects of SCAT7 on proliferation, apoptosis, and senescence reversed? Could the authors monitor a handful of endogenous HRNPK/YBX1 target genes after modulating SCAT7 levels, to solidify the idea that SCAT7 influences this transcriptional program?

This information is already shown in **Figs. 7d-g of the previous version of the manuscript**. Independent silencing of hnRNPk and YBX1 induces cell cycle perturbations, reduces the proliferation capacity of cancer cells and negatively affects the downstream oncogenic pathways such as PI3K/AKT and Ras/MAPK signaling pathways. As we mentioned in the discussion section, our results are consistent with the previous literatures that report the vital roles of hnRNPk and YBX1 in regulating a diverse array of biological functions. Moreover, our mechanistic study presented in **Fig. 8k** demonstrates a model of SCAT7-dependent regulation of FGFRs mediated through an

interaction with hnRNPK and YBX1. Thus, the *SCAT7*/hnRNPK/YBX1 complex promotes the cancer progression; however, perturbing any component of the complex negatively affects the cancer cell homeostasis.

As a follow up report of the current study, we have performed independent RNA-seq analyses of hnRNPK and YBX1 depleted cells to identify their transcriptome-wide target genes. We obtained a statistically significant overlap between *SCAT7* and hnRNPK target genes (398 genes). In case of YBX1 knockdown, we obtained 158 significant common genes. Since the latter study is a part of a follow-up detailed systematic study on the common functions of *SCAT7*/hnRNPK/YBX1 complex (other than regulation of cell cycle) and therefore wish to provide this information in a confidential manner.

3. The text reads well in general, but needs some editing to correct typos (e.g., ‘Principal Component Analysis’, not ‘Principle Component Analysis’) as well as for grammar and syntax.

In the revised manuscript, we have corrected the spelling mistakes and grammar mistakes.

Reviewer #3 (Remarks to the Author):

The authors profiled the S-phase specific lncRNAs and identified lots of candidates that are differentially expressed across cancer types and can potentially serve as prognostic biomarkers. Although the regulatory roles of lncRNAs in S-phase have not been fully investigated, this study provides a rich resource for our understanding of S-phase specific lncRNAs. Furthermore, the authors investigated the potential mechanism of a lncRNA, *SCAT7*, which may promote tumor progression by alternating signaling pathways relating to the hallmarks of cancer. The role of *SCAT7* in tumorigenesis was validated at molecular level, in multiple cell lines and through mouse xenografts. Several issues need to be addressed as follows.

Major points

1. In the first part of the Results section, the authors showed that the EtU labeling is efficient for isolation S-phase transcripts. However, in Figure 2B, the expression pattern of S phase-T2 specific lncRNAs is much more close to the expression pattern of unsynchronized cells compared to S-phase T1 and T3. Then in Figure 2C, the authors showed that 4 cluster of lncRNAs with distinctive temporal expression patterns across the S-phase T1, T2, T3 and unsynchronized cells. Have the authors investigated the functional characteristics of these 4 cluster or any biological processes that is related to the temporal expression pattern?

We assume that reviewer is addressing **Fig. 1c** not **Fig. 2c**. The four clusters presented in **Fig. 1c** represent the temporal patterns of lncRNAs (not protein coding genes). Hence, it was not possible to check the biological functions or perform an enrichment analysis for lncRNAs. For that reason, we looked for nearby protein coding genes (first hit to the S-phase lncRNA) and used those protein coding genes for gene enrichment analysis as already shown in **Fig. 1d**.

2. In the first step of experiment design, the authors selected 1,145 lncRNAs showing significant temporal expression patterns in HeLa cells across S-phase compared to unsynchronized cells as ‘S-phase specific’ lncRNAs. However, the TCGA samples are unsynchronized cells and may mostly consist of G1 phase cells according to the cell population in cancer cell lines in this study. Hence, lncRNAs that are significantly highly expressed in TCGA samples are not necessarily S-phase specific and may depend on the cancer type. These S-phase specific lncRNAs are a subset of expressed lncRNAs in HeLa cells and should not be considered as S-phase specific lncRNAs across

cancer types. It is better to show the difference between these S-phase specific lncRNAs and other differentially expressed lncRNAs across cancer types in terms of expression patterns and targets, etc.

We apologize for the confusing terminology. In the revised manuscript, we have changed the terminology from “S-phase specific” to “S-phase enriched” lncRNAs. However, in our manuscript, we do not state that the cancer associated lncRNAs are more specific to the S-phase. Rather, we identified lncRNAs that are enriched in the S-phase and used them as a resource to identify the differentially expressed lncRNAs in different cancer types. It is more possible that some of these S-phase identified lncRNAs may have expression in G1 or G2/M phase (we do not have sequencing samples to verify all these lncRNAs to be S-phase specific). At least the candidates we selected for functional validations (*SCAT1-8*) show more S-phase enrichment compared to G1 or G2/M phase as indicated by qPCR validation (Please see our response to the reviewer #1 comments one and two).

3. The Methods part of the paper is poorly written in the main text. Some of them are scattered in the supplementary materials. The authors need to describe them clearly in the paper, especially various methods used for data analyses.

We apologize for the concise description of the methods. In the revised manuscript we elaborated the methods part.

4. The data need to more carefully analyzed. For instance, to analyze the function of the S-phase specific lncRNAs, they used the neighboring protein-coding RNAs within 50kb window. The reason of 50Kb need to be explained. Different sizes of windows may be used to check if the enrichments of certain functional terms are robust.

We are sorry that we did not write the methods part in detail. In this study we have used first protein coding hit to the S-phase expressed lncRNAs within 50 kb. The 50 kb window was fixed because most (more than 85 %) of the first protein coding hit was within 50 kb range. We have also corrected the methods part to make clearer.

5. The authors used “data not shown” in multiple places, which should be avoided. Data, Tables and figures (maybe as supplementary materials) need to be provided.

In the revised manuscript, we have added some data as **Supplementary Fig. 5o** and **Supplementary Table 3j**.

Minor points

1. The authors mentioned bioinformatics tools such as Short Time-series Expression Miner (STEM) clustering and

GeneSCF. Maybe a brief introduction of the purpose and advantages of the tools and description of the results should be included to help the reader to understand the results.

Now the tools are properly introduced in the methods section. Also proper description for the columns from the GeneSCF output is now provided in the **Supplementary Table 1e-g** (Gene enrichment analysis for nearby protein coding genes).

2. Table S1 contains multiple tables of different categories and it is not clear which of Table S1 is referred to in the main text. Maybe a summary and a brief description should be included along with the tables or refer the tables in the Excel file as different numberings (e.g. Table S1, Table S2, Table S3 ...).

In the revised manuscript, we have included a brief description of different tables in the first sheet of each Excel file.

3. Why two bands in Figure7G of FGFR2 and ERK1/2? Are there some modifications?

The antibody used in this study identifies both p44/42 MAP kinase which gives two distinct bands according to the manufacturer datasheets.

4. In line 311, they declare that “H2228 and HepG2 cell lines were too sensitive to long term SCAT7 downregulation using shRNA particles.” Some supporting references and/or data are needed.

Since *SCAT7* functions are shown for the first time in the current manuscript, no literature will be found on *SCAT7*. In our *in vitro* experiments, the stable knockdown of *SCAT7* in H2228 and HepG2 cells reduced the viability drastically and eventually led to the death of all transduced cells. This observation in particular highlights the important role of *SCAT7* in cell proliferation.

REVIEWERS' COMMENTS:

Reviewer #1 (Remarks to the Author):

This is a much improved ms. I have a few minor comments.

1. SCAT7 RNA is induced during S-phase of the cell cycle. At the same time, SCAT7-depleted cells show G1 arrest. Depletion of an RNA that plays vital role in S-phase, should have shown defects in S phase progression and because of that cells should have accumulated in early S phase of the cell cycle. It is not clear to me why KD of an lncRNA critical for S phase progression results in G1 arrest.
2. Reduced number of EdU +ve cells upon SCAT7 depletion suggests that SCAT7-depleted cells do not reach S-phase. This phenotype does not mean that SCAT7 plays important role in S phase progression, as suggested by the authors. This only reflects less number of S phase cells in the population, presumably because of G1 arrest.
3. Authors should include flow cytometry data showing synchronization of HeLa cells using serum-starvation.

Reviewer #2 (Remarks to the Author):

The authors have addressed my concerns adequately.

Reviewer #3 (Remarks to the Author):

All my comments have been addressed well.

Response to Reviewer's comments

1. SCAT7 RNA is induced during S-phase of the cell cycle. At the same time, SCAT7-depleted cells show G1 arrest. Depletion of an RNA that plays vital role in S-phase, should have shown defects in S phase progression and because of that cells should have accumulated in early S phase of the cell cycle. It is not clear to me why KD of an lncRNA critical for S phase progression results in G1 arrest.

We appreciate the reviewer's concern regarding the functionality of *SCAT7* in the S-phase progression. In this context, according to the steady-state and nascent RNA sequencing coupled with qPCR validations, *SCAT7* is not exclusively expressed in the S-phase. However, it is expressed throughout the entire cell cycle stages with a significant enrichment in the S-phase (\log_2 FC = 1.47, 2.05 and 2.15 in early, mid and late S-phase stages, respectively) compared to unsynchronized cells. Consistent with its S-phase enrichment, *SCAT7* depletion in multiple cell lines exhibited a significant reduction (~50%) in the percentage of cells progressing through the S-phase which indicates a direct implication of *SCAT7* in the S-phase. One of the reasons for G1 cell accumulation in the *SCAT7* depleted cells could be due to significantly reduced levels of cyclin D1 (*CCND1*) at both transcriptional and translational levels as represented in **Figure 6e** accompanied with a significant hypophosphorylation of the retinoblastoma tumor suppressor gene (*Rb*). The combined effect of *CCND1* downregulation and *Rb* hypophosphorylation leads to a cell cycle arrest at the G1 phase that restricts the critical transition to the S-phase and hence negatively affects the S-phase progression (PMID:18719710, PMID: 18406353). Consistent with the *CCND1* and *Rb* deregulation, *SCAT7* knockdown altered the survival signaling of PI3K/AKT pathway which has been widely implicated in the downstream phosphorylation activity of the forkhead transcription factors (*FoxO1/3*) (PMID:10102273, PMID:10217147). The reduced AKT activity promotes the inhibitory effects of *FoxO1/3* factors leading to the suppression of type D cyclins and inhibition of the S-phase transition (PMID:12391153).

2. Reduced number of EdU +ve cells upon *SCAT7* depletion suggests that *SCAT7*-depleted cells do not reach S-phase. This phenotype does not mean that *SCAT7* plays important role in S phase progression, as suggested by the authors. This only reflects less number of S phase cells in the population, presumably because of G1 arrest.

In the current manuscript, the ultimate necessity of *SCAT7* expression for the maintenance of S-phase population has been assessed using both flow-cytometry and EdU incorporation assay. The drastic reduction in the EdU +ve cells is a clear indication of the diminished S-phase within DNA-replicating cells, indicating a possible role for *SCAT7* in S-phase progression. We agree with the reviewer's opinion that the reduction in the S-phase population may be a consequence of the G1 arrest and more work is needed to ascertain the functions of *SCAT7* in S-phase

progression. **[Editorial Note: Unpublished data redacted from Peer Review File as per Authorial Request.]**

3. Authors should include flow cytometry data showing synchronization of HeLa cells using serum-starvation.

We included the representative data as Supplementary Figure 1e.